# Parallelizing MCMC Across the Sequence Length

**David M. Zoltowski**[*]
Stanford University
dzoltow@stanford.edu

**Skyler Wu**[*]
Stanford University
skylerw@stanford.edu

**Xavier Gonzalez**
Stanford University
xavier18@stanford.edu

**Leo Kozachkov**
Brown University
leokoz8@brown.edu

**Scott W. Linderman**
Stanford University
scott.linderman@stanford.edu

## Abstract

Markov chain Monte Carlo (MCMC) methods are foundational algorithms for Bayesian inference and probabilistic modeling. However, most MCMC algorithms are inherently sequential and their time complexity scales linearly with the sequence length. Previous work on adapting MCMC to modern hardware has therefore focused on running many independent chains in parallel. Here, we take an alternative approach: we propose algorithms to evaluate MCMC samplers *in parallel across the chain length*. To do this, we build on recent methods for parallel evaluation of nonlinear recursions that formulate the state sequence as a solution to a fixed-point problem and solve for the fixed-point using a parallel form of Newton's method. We show how this approach can be used to parallelize Gibbs, Metropolis-adjusted Langevin, and Hamiltonian Monte Carlo sampling across the sequence length. In several examples, we demonstrate the simulation of up to hundreds of thousands of MCMC samples with only tens of parallel Newton iterations. Additionally, we develop two new parallel quasi-Newton methods to evaluate nonlinear recursions with lower memory costs and reduced runtime. We find that the proposed parallel algorithms accelerate MCMC sampling across multiple examples, in some cases by more than an order of magnitude compared to sequential evaluation.

## 1 Introduction

Markov chain Monte Carlo (MCMC) algorithms sequentially generate samples from a target distribution, with computational cost linear in the length of the chain [1–3]. While modern hardware like GPUs and TPUs can dramatically accelerate parallelizable algorithms, leveraging these advances for MCMC remains challenging. In particular, while parallel resources can be used to simulate many independent chains in parallel [4, 5], the runtime remains linear in the sequence length.

In this paper, we present a promising approach for parallelizing MCMC algorithms across the sequence length to obtain sublinear chain-length complexity. The core idea is to adapt methods for the parallel evaluation of nonlinear recursions [6–8] to MCMC algorithms. These methods formulate the state sequence of a nonlinear sequence model as the solution of an optimization problem and iteratively solve for the state sequence via a parallel-in-time formulation of Newton's method.

We show that these methods can parallelize widely used classes of MCMC samplers across the chain length, including Gibbs sampling [9], the Metropolis-adjusted Langevin algorithm (MALA) [10], and Hamiltonian Monte Carlo (HMC) [11–13]. Across these examples, we often find that dozens of parallel evaluations are sufficient to generate anywhere from thousands to hundreds of thousands of samples. We also propose two new variations of the parallel Newton method that reduce computational costs and improve efficiency for certain MCMC applications. With these approaches, we find that the proposed parallel algorithms can dramatically accelerate MCMC sampling.

---

[*]Equal contribution.

39th Conference on Neural Information Processing Systems (NeurIPS 2025).

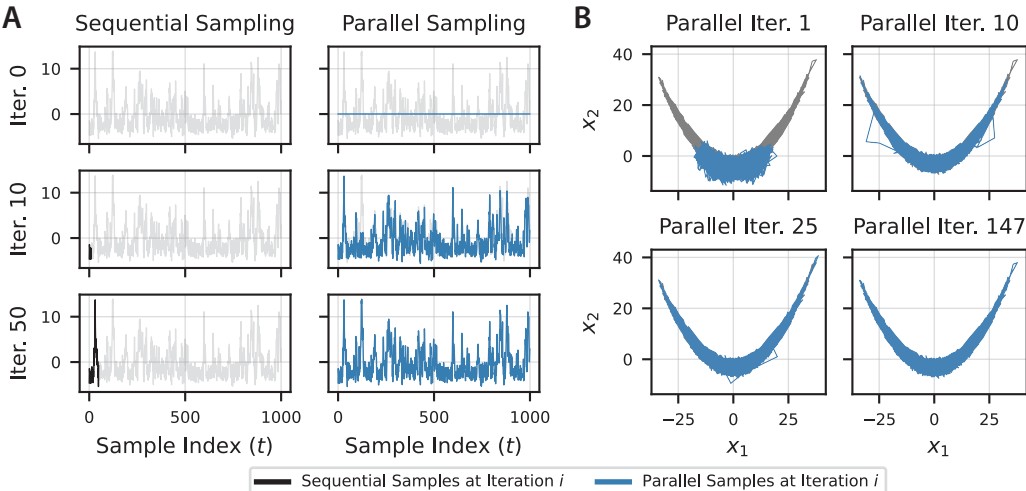

Figure 1: **Parallel evaluation of 100K HMC samples targeting the Rosenbrock distribution. (A)** Dark lines (black: sequential, blue: parallel) show simulated samples of the second coordinate for the first 1K samples at three iterations of each method. The light gray line is the final sequential chain. **(B)** 100K samples after 1, 10, 25, or 147 parallel iterations (blue), converging to the sequential trace (gray) by iteration 147.

The paper is organized as follows. We first review how Newton's method can parallelize nonlinear recursions, then show how this framework applies to Gibbs sampling, MALA, and HMC across the sequence length. For HMC, we additionally propose parallelizing only the leapfrog integration within each proposal. Next, we introduce scalability improvements to parallel Newton methods and a novel block quasi-Newton variant adapted for parallelizing leapfrog integration. Finally, we present experiments demonstrating wall-clock speedups of parallel MCMC sampling across multiple examples. In Figure 1, we illustrate our approach by simulating 100K HMC samples from a high-curvature 2D distribution, where parallel HMC converges to the sequential trace in just 147 steps. Our implementation is available at https://github.com/lindermanlab/parallel-mcmc.

## 2   Background

Our work builds on and extends previous approaches for parallelizing linear and nonlinear recursions. Consider a recursion, $\mathbf{s}_t = f_t(\mathbf{s}_{t-1})$, that when given an initial state $\mathbf{s}_0$ yields a sequence of states, $\mathbf{s}_{1:T} = (\mathbf{s}_1, \ldots, \mathbf{s}_T) \subset \mathbb{R}^D$, according to the transition functions, $f_t : \mathbb{R}^D \mapsto \mathbb{R}^D$. The obvious way to solve for the sequence of states is to evaluate the recursion sequentially. However, if $f_t$ is an affine function (i.e., $f_t(\mathbf{s}_{t-1}) = \mathbf{J}_t \mathbf{s}_t + \mathbf{u}_t$ for some matrix $\mathbf{J}_t$ and vector $\mathbf{u}_t$), then the system can be evaluated in $\mathcal{O}(\log T)$ time on a parallel machine using a parallel (a.k.a. associative) scan [14]. The key insight is that a composition of updates, $f_{t+1} \circ f_t$, is still an affine function. With $\mathcal{O}(T)$ processors, one can compute all pairs of updates in parallel, yielding a sequence that is half as long. By repeating this process $\mathcal{O}(\log T)$ times, one can obtain the entire sequence in sublinear time. This parallel algorithm underlies many modern machine learning models for sequential data [15–17].

In contrast, nonlinear sequential processes such as recurrent neural networks and, for our purposes, common MCMC algorithms, are not directly parallelizable and instead are typically evaluated sequentially. However, Danieli et al. [6] and Lim et al. [7] showed that we can view the states of a nonlinear recursion as the solution of a fixed-point equation, and that a parallelized form of Newton's method can be used to solve for the state sequence. Lim et al. [7] and Danieli et al. [6] observed that iteratively linearizing the transition functions, $f_t$, around a guess for the state trajectory, $\mathbf{s}_{1:T}^{(i)}$, and evaluating the resulting linear system to obtain a new trajectory, $\mathbf{s}_{1:T}^{(i+1)}$, is equivalent to using Newton's method on an appropriately defined residual. More precisely, given $\mathbf{s}_0$ and $f_{1:T}$, the residual is the vector of temporal differences

$$\mathbf{r}(\mathbf{s}_{1:T}) = \text{vec}([\mathbf{s}_1 - f_1(\mathbf{s}_0), \ldots, \mathbf{s}_T - f_T(\mathbf{s}_{T-1})]) \tag{1}$$

and the Newton update is given by the Taylor expansion,

$$\mathbf{s}_t^{(i+1)} = f_t(\mathbf{s}_{t-1}^{(i)}) + \mathbf{J}_t \left( \mathbf{s}_{t-1}^{(i+1)} - \mathbf{s}_{t-1}^{(i)} \right), \quad \text{where} \quad \mathbf{J}_t := \frac{\partial f_t}{\partial \mathbf{s}}(\mathbf{s}_{t-1}^{(i)}). \tag{2}$$

By rearranging terms it is clear this is a linear dynamical system

$$\mathbf{s}_t^{(i+1)} = \boldsymbol{J}_t \mathbf{s}_{t-1}^{(i+1)} + \underbrace{f_t(\mathbf{s}_{t-1}^{(i)}) - \boldsymbol{J}_t \mathbf{s}_{t-1}^{(i)}}_{\mathbf{u}_t} \tag{3}$$

with dynamics given by the time-varying Jacobians, $\boldsymbol{J}_t$, and inputs, $\mathbf{u}_t$. The Jacobians and inputs only depend on the states from the previous Newton iteration, and thus can be computed in parallel. Crucially, this linear dynamical system can also be solved in $\mathcal{O}(\log T)$ time on a parallel machine using the parallel scan algorithm described above. This parallel update is repeated until the state sequence converges within a numerical tolerance $\delta$. This algorithm was termed "DEER" by Lim et al. [7] and "DeepPCR" by Danieli et al. [6]. For clarity, we refer to it as "DEER" or "Newton's method."

The DEER algorithm requires computing, storing, and multiplying $T$ Jacobian matrices $\boldsymbol{J}_t$ of size $D \times D$. Therefore, DEER must perform $\mathcal{O}(TD^3)$ work and requires $\mathcal{O}(TD^2)$ memory, which can be prohibitive in larger dimensionalities. To reduce computational complexity, Gonzalez et al. [8] proposed quasi-DEER, a quasi-Newton method that retains only the diagonal of the Jacobian matrices, $\mathrm{diag}(\boldsymbol{J}_t)$, and evaluates the following linear recursion

$$\mathbf{s}_t^{(i+1)} = \mathrm{diag}(\boldsymbol{J}_t)\,\mathbf{s}_{t-1}^{(i+1)} + f_t(\mathbf{s}_{t-1}^{(i)}) - \mathrm{diag}(\boldsymbol{J}_t)\,\mathbf{s}_{t-1}^{(i)}. \tag{4}$$

This reduces storage and matrix multiplication costs to $\mathcal{O}(TD)$, which generally improves wall-clock time and memory usage compared to DEER. However, computing the diagonal of the Jacobian via automatic differentiation still requires $D$ passes through the function, which can be slow or memory-intensive. Gonzalez et al. [8] thus propose hard-coding the diagonal of the Jacobian when possible. In Section 3.4, we propose an alternative that requires only a single forward pass through the function and does not require hard-coded diagonal Jacobians. In summary, DEER variants evaluate nonlinear sequence models by iteratively refining an initial state sequence using parallel-in-time updates. A high-level overview of this procedure is shown in Algorithm 1 in Appendix A.

# 3 Parallelizing MCMC

In this work, we leverage parallel Newton methods to evaluate MCMC samplers. We set the sequence of functions $f_{1:T}$ to be the transitions of an MCMC sampler and $\mathbf{s}_0$ to be the initial state of the chain. We then iteratively solve for the resulting MCMC sampling sequence $\mathbf{s}_{1:T}$. In this section we describe the details of this approach for parallelizing reparameterized Gibbs sampling, MALA, and HMC, as well as several techniques for improving the efficiency and scalability of parallel MCMC.

## 3.1 Parallel reparameterized Gibbs sampling

Gibbs samplers partition the joint distribution of the target into conditional distributions that can be alternately sampled [9]. Here we focus on a subset of Gibbs samplers in which each conditional distribution is reparameterizable, such that it is a deterministic and differentiable function of input randomness. Suppose we are targeting a distribution $p(\boldsymbol{x})$ over a random variable $\boldsymbol{x} \in \mathbb{R}^D$. Let $\boldsymbol{x}_t = (x_{t,1}, \ldots, x_{t,D}) \in \mathbb{R}^D$ denote the state of the Gibbs sampler at iteration $t$. We define $f$ to be the Gibbs sweep that iteratively updates each coordinate given input noise, $\boldsymbol{\xi}_t \in \mathbb{R}^D$, such that

$$\boldsymbol{x}_t = f(\boldsymbol{x}_{t-1}, \boldsymbol{\xi}_t) \triangleq f_t(\boldsymbol{x}_{t-1}). \tag{5}$$

Even though the MCMC transition kernel is fixed, the reparameterized update functions vary with time because the input noise, $\boldsymbol{\xi}_t$, changes at each step. More specifically, let $f_d$ be the Gibbs update of coordinate $d$ conditioned on all other coordinates,

$$x_{t,d} = f_d\big((x_{t,1}, \ldots, x_{t,d-1}, x_{t-1,d+1}, \ldots, x_{t-1,D}), \xi_{t,d}\big), \tag{6}$$

which importantly does not depend on $x_{t-1,d}$. Then $f$ is the composition of the $D$ coordinate updates $f = f_1 \circ f_2 \circ \cdots \circ f_D$. Observe that when put in the form of equation (5) this is a nonlinear dynamical system with states, $\mathbf{s}_t = \boldsymbol{x}_t$, and transition functions, $f_t$, which vary in time due to the input randomness $\boldsymbol{\xi}_t$. Thus, we can directly apply the DEER algorithm to parallelize sampling.

## 3.2 Metropolis-adjusted Langevin dynamics (MALA)

MALA generates proposals via Langevin dynamics and then accepts or rejects the proposed state with a Metropolis correction [10]. For target distribution $p(\mathbf{x})$ and step size $\epsilon$, the proposal is generated via

$$\tilde{\mathbf{x}}_t = \mathbf{x}_{t-1} + \epsilon \nabla_{\mathbf{x}} \log p(\mathbf{x}_{t-1}) + \sqrt{2\epsilon}\,\boldsymbol{\xi}_t, \quad \boldsymbol{\xi}_t \sim \mathcal{N}(\mathbf{0}, \boldsymbol{I}) \tag{7}$$

and accepted with probability $\min\{1, p(\tilde{\mathbf{x}}_t)q(\mathbf{x}_{t-1} \mid \tilde{\mathbf{x}}_t)/p(\mathbf{x}_{t-1})q(\tilde{\mathbf{x}}_t \mid \mathbf{x}_{t-1})\}$ where $q(\tilde{\mathbf{x}}_t \mid \mathbf{x}_{t-1})$ is the proposal density of candidate $\tilde{\mathbf{x}}_t$ given $\mathbf{x}_{t-1}$. Importantly, MALA can be cast as a nonlinear recursion amenable to parallelization by setting the state $\mathbf{s}_{t-1} = \mathbf{x}_{t-1}$. The MALA update function $f_t(\boldsymbol{x}_{t-1}) \triangleq \mathrm{MALA}(\boldsymbol{x}_{t-1}, \{\boldsymbol{\xi}_t, u_t\})$ depends on input random variables $\boldsymbol{\xi}_t \sim \mathcal{N}(\mathbf{0}, \boldsymbol{I})$ and $u_t \sim \mathcal{U}(0, 1)$. We define it as follows, with logistic function $\sigma$ and binary indicator function $\mathbb{1}(\ldots)$:

> **function** MALA($\mathbf{x}_{t-1}, \{\boldsymbol{\xi}_t, u_t\}$)
> $\quad \tilde{\mathbf{x}}_t \leftarrow \mathbf{x}_{t-1} + \epsilon \nabla_{\mathbf{x}} \log p(\mathbf{x}_{t-1}) + \sqrt{2\epsilon}\, \boldsymbol{\xi}_t$
> $\quad \alpha \leftarrow \min\{1, p(\tilde{\mathbf{x}}_t)q(\mathbf{x}_{t-1} \mid \tilde{\mathbf{x}}_t)/p(\mathbf{x}_{t-1})q(\tilde{\mathbf{x}}_t \mid \mathbf{x}_{t-1})\}$
> $\quad \tilde{g} \leftarrow \log \alpha - \log u_t$
> $\quad g \leftarrow \sigma(\tilde{g}) + \texttt{stop\_gradient}(\mathbb{1}(\tilde{g} > 0) - \sigma(\tilde{g}))$       ▷ stop-gradient trick
> $\quad \mathbf{x}_t \leftarrow g\, \tilde{\mathbf{x}}_t + (1 - g)\, \mathbf{x}_{t-1}$         ▷ accept-reject via gating variable
> $\quad$ **return** $\mathbf{x}_t$
> **end function**

Notably, the accept–reject step renders the MALA update non-differentiable. In our proposed MALA function, the exact MALA update is computed in the forward pass, while the Jacobian for DEER is obtained using the stop-gradient trick [see 18], which yields a differentiable relaxation of the accept–reject step. Critically, Proposition 1 of Gonzalez et al. [8] guarantees that DEER and quasi-DEER still globally converge to the true sequential MALA trace in this setting, despite the approximate Jacobian and non-differentiable $f_t$. That is, given shared input randomness, both parallel and classical sequential MALA return the same set of samples up to the numerical tolerance $\delta$.

### 3.3 Parallelizing Hamiltonian Monte Carlo

Hamiltonian Monte Carlo (HMC) is a prominent MCMC algorithm that uses gradient information with augmented momentum variables to efficiently explore the sampling space [11–13]. Each step of HMC consists of sampling new momenta, integrating Hamiltonian dynamics, and accepting or rejecting the proposed state. In the following, we present two approaches for parallelizing HMC that either: 1) parallelize across sampling steps; or 2) parallelize the leapfrog integration within each step. We focus on HMC with a fixed number of leapfrog steps and step size and discuss considerations for future work on parallelizing NUTS [19] and other adaptive HMC algorithms in Section 6.

**Parallel HMC with Sequential Leapfrog Integration.** The most straightforward way to parallelize HMC is to extend our MALA strategy and parallelize across HMC sampling steps. Here, the function $f_t$ runs a *sequential* leapfrog integrator to generate a new proposal. As before for MALA, we substitute a differentiable relaxation of the accept-reject step for the Jacobian. We used this method in Figure 1 and apply it to logistic regression in Section 5.

**Sequential HMC with Parallel Leapfrog Integration.** Alternatively, we could run HMC steps sequentially and parallelize the inner leapfrog integration loop. Let $p(\mathbf{x})$ be the target distribution with $\mathbf{x} \in \mathbb{R}^D$. Each HMC step samples a momentum variable $\mathbf{v} \in \mathbb{R}^D$ and then integrates Hamiltonian dynamics for $L$ steps using the leapfrog integrator with step size $\epsilon$. If the momentum is updated with a half-step before integrating, the resulting integration step is

$$\mathbf{x}_t = \mathbf{x}_{t-1} + \epsilon\, \mathbf{v}_{t-1}; \quad \mathbf{v}_t = \mathbf{v}_{t-1} + \epsilon\, \nabla_{\mathbf{x}} \log p(\mathbf{x}_t) \tag{8}$$

for $t = \{1, ..., L\}$. After integration, the momentum is updated with a negative half-step and then the candidate state is accepted or rejected. To apply DEER to this problem, we set the state to the concatenation of the position and momentum states $\mathbf{s}_t = [\mathbf{x}_t, \mathbf{v}_t]$ and define $f_t : \mathbb{R}^{2D} \to \mathbb{R}^{2D}$ as the leapfrog integrator steps in equation (8). The complete algorithm is presented in Appendix A.2. This novel ability to parallelize leapfrog integration raises new considerations about the optimal step size and acceptance rates, which we discuss in Appendix B.3.

**Block quasi-DEER for parallelizing leapfrog integration.** The Jacobian of the leapfrog step has the following block structure, where $\boldsymbol{I}_D$ is an identity matrix:

$$\boldsymbol{J}(\mathbf{s}_{t-1}) = \frac{\partial f}{\partial \mathbf{s}}(\mathbf{s}_{t-1}) = \begin{bmatrix} \boldsymbol{I}_D & \epsilon \boldsymbol{I}_D \\ \epsilon \nabla_{\mathbf{x}}^2 \log p(\mathbf{x}_{t-1} + \epsilon \mathbf{v}_{t-1}) & \boldsymbol{I}_D + \epsilon^2 \nabla_{\mathbf{x}}^2 \log p(\mathbf{x}_{t-1} + \epsilon \mathbf{v}_{t-1}) \end{bmatrix}. \tag{9}$$

The diagonal quasi-DEER algorithm discards information in the off-diagonal blocks. We therefore propose an alternative "block quasi-DEER" algorithm that employs the diagonal of each *block* of the Jacobian. For leapfrog integration, this approximate Jacobian is exact for the top row and takes into account interactions between the position and momentum states in the off-diagonal blocks. Crucially, the block quasi-DEER matrix form can also be parallelized via an efficient linear recursion with $\mathcal{O}(TD)$ memory and $\mathcal{O}(TD)$ work. See Appendix A for additional details of this approach.

### 3.4 Scaling parallel MCMC

In this section, we propose several ways to make parallel MCMC more efficient by reducing the computational cost of each update or reducing the number of parallel Newton iterations.

**A general stochastic approach for efficient quasi-DEER.**    Quasi-DEER [8] and our block variant require the diagonal of a Jacobian, which is not always easily available in closed form. Computing the diagonal via automatic differentiation requires $D$ passes through the function to compute the full Jacobian and retain only the diagonal elements, which is computationally and memory intensive for large systems. We therefore propose a stochastic quasi-DEER that is generally memory efficient and still enjoys global convergence guarantees due to Proposition 1 of Gonzalez et al. [8]. Our approach leverages a general stochastic estimator of the diagonal of the Jacobian [20, 21] based on Hutchinson's method [22] and given by the following, where each element $\mathbf{z}_i \sim$ Rademacher:

$$\mathrm{diag}(\boldsymbol{J}_t) = \mathbb{E}_{\mathbf{z} \sim \mathrm{Rad}}[\mathbf{z} \odot (\boldsymbol{J}_t \mathbf{z})]. \tag{10}$$

This expectation can be estimated via standard Monte Carlo using efficient Jacobian-vector products. The resulting stochastic quasi-DEER algorithm joins the family of memory-efficient quasi-DEER algorithms proposed in Gonzalez et al. [8]. Empirically, we find that only one or a few Monte Carlo samples often suffice to estimate $\mathrm{diag}(\boldsymbol{J}_t)$. Notably, when using a single sample estimate the stochastic quasi-DEER algorithm requires only one forward pass through the function $f_t$ to evaluate it at the state $\mathbf{s}_{t-1}$ and compute the Jacobian-vector product $\boldsymbol{J}_t \mathbf{z}$. This substantially improves over the $D$ passes through the function required for automatic differentiation and matches the number of forward passes required by Picard iterations [23]. In our results, stochastic quasi-DEER yields major efficiency gains, and we expect it to be broadly useful for parallel evaluation of nonlinear recursions.

**High quality approximate samples with early-stopping.**    The sample sequence generated via parallel Newton iterations at convergence is equal to the sequence generated from sequential evaluation, up to numerical tolerance. However, we observe that intermediate sample sequences before convergence can still apparently produce high-quality samples. For example, consider the samples generated at the 25th parallel iteration in Figure 1. We therefore propose an approximate sampling scheme where samples are generated by early-stopping the parallel Newton iterations before convergence.

**Orthogonal coordinate transformation for quasi-DEER.**    As quasi-DEER relies on a diagonal approximation to the Jacobian, its performance can depend on the accuracy of this approximation. To mitigate this issue in some problems, we propose to reparameterize the system for quasi-DEER using an orthogonal coordinate transformation $\mathbf{z}_t = \boldsymbol{Q}^\top \mathbf{s}_t$ with $\boldsymbol{Q} \in \mathbb{R}^{D \times D}$ and $\boldsymbol{Q}^\top \boldsymbol{Q} = \boldsymbol{I}$. In this new set of coordinates, the dynamics of the system are given by

$$\mathbf{z}_t = \boldsymbol{Q}^\top f_t(\boldsymbol{Q}\mathbf{z}_{t-1}) \triangleq \hat{f}_t(\mathbf{z}_{t-1}). \tag{11}$$

The resulting Jacobian in these coordinates is $\hat{\boldsymbol{J}}_t = \boldsymbol{Q}^\top \boldsymbol{J}_t \boldsymbol{Q}$. If $\hat{\boldsymbol{J}}_t$ is closer to diagonal than $\boldsymbol{J}_t$, the quasi-DEER approximation will be more accurate. At convergence, we map the transformed states back to the original coordinates via $\mathbf{s}_t = \boldsymbol{Q}\mathbf{z}_t$. These efforts to make Markovian systems more amenable to quasi-DEER are part of a broader theme in the community; for example, Farsang et al. [24] develops a nonlinear RNN where the dynamics are constrained to be diagonal. The orthogonal change of coordinates is particularly effective for dynamical systems with real and symmetric (or near-symmetric) Jacobians. Importantly, this is the case for MALA. If we ignore the accept-reject step in MALA then the Jacobian is $\boldsymbol{J}_t = \boldsymbol{I} + \epsilon \nabla_{\mathbf{x}}^2 \log p(\mathbf{x}_{t-1})$. Assuming continuity of second derivatives of $\log p(\mathbf{x})$, this is a real and symmetric matrix that can be diagonalized with an orthogonal matrix $\boldsymbol{Q}$.

**Sliding window updates.**    To improve algorithm performance for higher-dimensional problems, we adapted the sliding window technique proposed in Shih et al. [23]. In this approach, at each iteration we apply a DEER update within a window of the sequence rather than across the entire sequence length. The window is initialized to start at the first time point. After each iteration, the window is shifted forward in the sequence to the first time point that has not yet converged within the tolerance.

## 4 Related Work

**Parallelizing diffusions.**    A related line of work has developed parallelizable algorithms for sampling from diffusions. Shih et al. [23] used Picard iterations, which also cast the state sequence as a fixed point, to parallelize diffusion model sampling. Follow-up works include the Parareal algorithm [25] by Selvam et al. [26] and Anderson acceleration via triangular nonlinear equations by Tang et al. [27]. In fact, recent work has shown that in this setting, Picard iterations can be interpreted

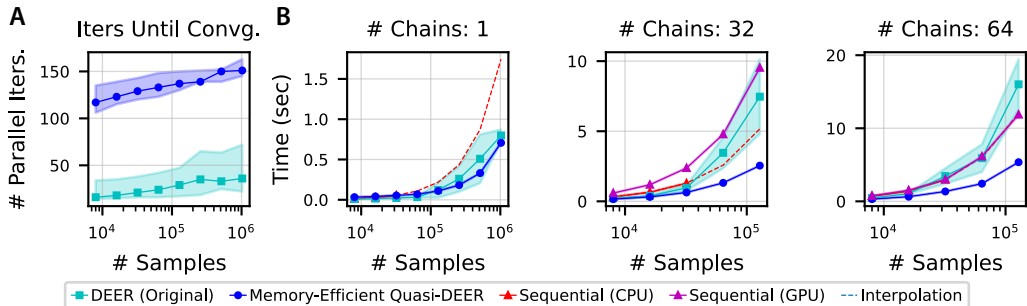

Figure 2: **Parallel Gibbs sampling.** **(A)** The number of iterations for DEER and preconditioned memory-efficient quasi-DEER scales approximately logarithmically in the sequence length (solid line is median and shaded areas are 90% confidence intervals). **(B)** Quasi-DEER is faster than sequential sampling on CPU or GPU across varying batch sizes, i.e., number of independent chains evaluated in parallel.

as a version of quasi-DEER where the Jacobian is replaced by the identity [28]. More closely related to our work, Anari et al. [29] used Picard iterations to parallelize Langevin diffusion sampling. While these studies are highly relevant, there are important differences from our work. First, they focus on parallel simulation of SDEs, which differ from MCMC algorithms involving discrete accept–reject steps and coordinate-wise updates. Next, Picard iterations achieve linear convergence, whereas our Newton-based approach attains quadratic convergence under certain conditions. Separately, Danieli et al. [6] further demonstrated the promise of parallel Newton methods for diffusion model simulation, showing theoretical speedups under the assumption of perfect Jacobian parallelization, though limited in practice by memory constraints. The stochastic quasi-DEER approach may alleviate such memory issues and aid parallelizing diffusion model sampling.

**Related work for parallel MCMC.** Many works have sought to accelerate MCMC via parallel computing [30, 4]. The most related is Grazzi and Zanella [31], which parallelized random-walk Metropolis sampling using Picard iterations. In contrast, we focus on Gibbs sampling and gradient-based MCMC algorithms and employ first-order Newton methods for parallelization. Beyond this, the most common approach is to run multiple independent chains in parallel, which is supported by modern MCMC libraries [32–35] and theoretical analysis [36, 5]. However, some samplers like the No-U-Turn Sampler (NUTS), an adaptive HMC algorithm [19] with additional control-flow computations that complicate GPU acceleration, may not be as easily parallelizable across chains. This motivated GPU-friendly adaptive HMC variants [37, 4] and finite-state machine formulations of MCMC algorithms that alleviate GPU synchronization issues [38]. Finally, many parallel-chain MCMC algorithms also share information across chains [39, 40].

Parallel computation in MCMC extends beyond batching independent chains. For Gibbs sampling, extensive work has focused on parallelizing Gibbs update sequences within a sample iteration [41–43]. In multi-proposal MCMC, multiple candidate states are generated in parallel to enhance target exploration [44–49]. Pre-fetching approaches parallelize tasks such as proposal generation alongside other computations [50, 51, 30]. Parallel-in-time algorithms have been developed for Bayesian smoothing in state space models [52–54] and are related to our work. Finally, many approaches partition data into subsets and run parallelizable inference processes for each data subset [55–57, 30].

# 5 Results

Our experiments evaluate the number of Newton iterations to convergence, wall-clock time relative to sequential sampling, and sample quality for parallel Gibbs, MALA, and HMC across multiple problems. Our implementations were in JAX [58] with wall-clock times measured post-JIT compilation. Unless otherwise noted, all runs used a single H100 GPU on a SLURM cluster.

## 5.1 Parallel Gibbs sampling for a hierarchical Gaussian model

We first demonstrate parallelization of a reparameterized Gibbs sampler for the eight schools problem [3, 59], a hierarchical Gaussian model of test scores $x_{s,n}$ for school $s$ and student $n$ with 20 students per school (see Appendix B.2 for additional details). We sampled synthetic data from this model with specified means and standard deviations per school, and used a parallelized Gibbs sampler to draw samples from the 18-dimensional posterior distribution.

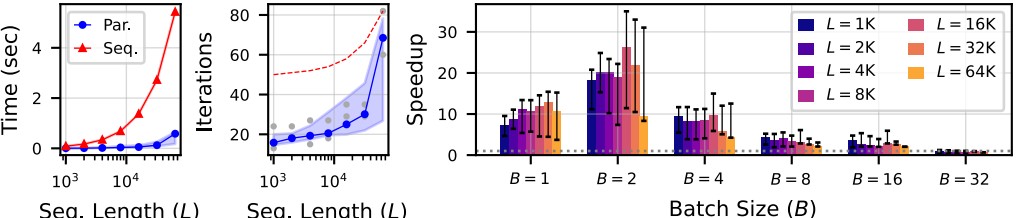

Figure 3: **Parallel MALA performance using efficient quasi-DEER (Q-DEER).** **(A)** Wall-clock times to sample chains of length $L$ for batch size $B = 2$ for parallel MALA ("Par.") vs. sequential MALA ("Seq."). **(B)** For batch size $B = 2$, the distribution/number of Newton iterations needed for each chain to numerically converge. The light-blue band represents a $60\%$ confidence interval over 20x random seeds. The gray dots show the number of iterations needed for numerical convergence of a particular seed as an example. The dashed line represents the maximum permitted number of iterations at a given setting of $L$. Parallel MALA often converges in only a few dozen iterations, even when generating tens of thousands of samples. **(C)** Wall-clock speedup multipliers of parallel over sequential MALA with $90\%$ confidence intervals over 20x random seeds. The gray dotted line marks a multiplier of 1.0, i.e. equal performance. For many batch sizes, Q-DEER can generate numerically-equivalent samples more than an order of magnitude faster than sequential sampling.

We investigated the convergence and wall-clock speed of DEER and efficient quasi-DEER on an A100 GPU using a 3-sample stochastic Jacobian diagonal estimate and diagonal preconditioning. We compared wall-clock times to sequential sampling baselines on GPU and CPU across various numbers of chains and chain lengths. We found parallel Newton methods converged rapidly for this problem, requiring tens of DEER iterations or 100-150 quasi-DEER iterations for chain lengths up to 1M samples (Figure 2A). Furthermore, for batches of 32 or 64 chains, Quasi-DEER yielded the fastest wall-clock sampling times across all settings (Figure 2B), achieving approximately $2\times$ faster sampling than sequential methods.

## 5.2 Parallel MALA for Bayesian Logistic Regression

We next evaluated parallel MALA, focusing on (1) the convergence rate of Newton's method and the wall-clock time relative to sequential MALA, and (2) its efficiency in generating useful samples. For parallel MALA, we used the stochastic quasi-DEER algorithm with a 1-sample estimate of the diagonal. We targeted the posterior of a Bayesian logistic regression (BLR) model of the German Credit Dataset with whitened covariates [60, 61] and a $\mathcal{N}(\mathbf{0}, \mathbf{I})$ prior. We compared parallel and sequential MALA with step size $\epsilon = 0.0015$ ($\approx 80\%$ acceptance rate) across batches of independent chains $B \in 1, 2, 4, 8, 16, 32$ and chain lengths $L \in 1K, 2K, 4K, 8K, 16K, 32K, 64K$. Each configuration was repeated across 20 random seeds. The main text reports representative subsets of $B$ and $L$ highlighting key trends; full results and metric details are in Appendix C.1.

**Wall-Clock Time and Convergence.** From Figures 3A and 3C, we observe that for all tested sequence lengths $L$ and batch sizes $B$ except for $B = 32$, parallel MALA can generate the same $B$ batches of $L$ samples each much faster than sequential MALA, achieving speedups of up to 20 or 30 times for some smaller batch sizes. However, sequential MALA is slightly faster than parallel MALA for batch size $B = 32$. This reflects a trade-off in the allocation of parallel resources, where for large batch sizes parallel MALA saturates our GPU resources, degrading performance. Eventually, parallel MALA resulted in out-of-memory errors with $B = 32$ and $L = 64K$, and scaling to this size of samples or larger with parallel MALA would require the sliding window technique. Notably, parallel MALA typically converged in tens of Newton iterations (fig. 3B), and only rarely had chains not yet converged at our prespecified `max_iters` (red dashed line in Figure 3B and Appendix B).

**On Sample Quality via MMD and Newly-Accessible Tradeoffs.** We next investigated the quality of samples generated via parallel MALA and whether parallel or sequential sampling was more efficient at generating high quality samples. To assess sample quality, we computed the Maximum Mean Discrepancy (MMD) [62] between sets of MALA samples and a ground truth set of samples computed via NUTS (Hoffman et al. [19], see Appendix B). This allowed us to compare sample quality across batch sizes and chain length with lower MMDs value implying higher sample quality.

Parallel MALA produced samples of virtually identical quality to sequential sampling as measured by MMD (Figure 4), while achieving over an order-of-magnitude speedup for smaller batch sizes.

For the largest batch size ($B = 32$), however, parallel MALA was slower than sequential MALA. To determine which combination of $B$, $L$, and sampling mode most efficiently generated useful samples, we computed *interpolated* timing estimates to identify the fastest configuration for simulating $B$ chains of $L$ samples. The purple stars in Figure 4 denote such interpolated settings: for instance, rather than one run with $B = 32$ and $L = 16K$, four parallel MALA runs with $B = 8$ and $L = 16K$ will likely achieve comparable sample quality in less time, while avoiding GPU oversaturation.

Across nearly all batch sizes and sequence lengths in Figure 4, sequentially-run interpolated parallel MALA (purple stars) lies below and to the left of sequential MALA (red triangles), indicating faster wall-clock times with equal or better sample quality. This result highlights a new tradeoff in GPU resource allocation of parallelizing across chain length versus batching independent chains. Our findings suggest that parallel MALA attains optimal performance when more resources are allocated for chain-length parallelization.

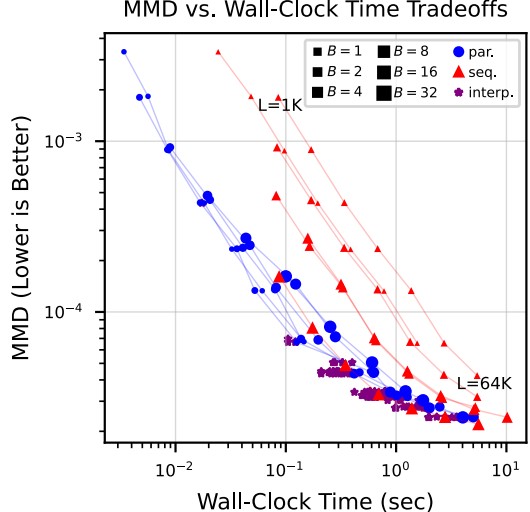

Figure 4: **MMD vs. wall-clock time performance for parallel and sequential MALA.** Blue circles denote parallel MALA, red triangles denote sequential MALA, and purple stars denote interpolated parallel MALA (e.g., sequentially running two independent $B = 4$, $L = 16K$ instances instead of one $B = 8$, $L = 16K$ run). Each line connects points of equal batch size across varying sequence lengths $L$. *Points closer to the bottom-left indicate greater compute-time efficiency (i.e., faster and higher quality).*

## 5.3 Parallel HMC

We next evaluated the parallel HMC algorithms for simulating samples from the same BLR model of the German Credit dataset. First, we compared the iterations until convergence for parallelizing leapfrog integration using DEER, quasi-DEER, and block quasi-DEER with $L = 32$ leapfrog steps across a range of step sizes and initial conditions (Figure 5A). We found that DEER converged in the fewest iterations, although it incurs a very high memory cost for this problem. Quasi-DEER incurs a light memory cost, but in this example often required close to the maximum number of parallel iterations to converge ($L$ iterations). Block quasi-DEER, which accounts for position and momentum interactions, notably reduced the number of iterations to convergence by $\approx 2\times$ compared to quasi-DEER across a range of relevant step sizes. In our timing experiments, we therefore solely considered block quasi-DEER for parallelizing the leapfrog integration.

We compared the time to generate 1000 HMC samples from the BLR model across varying numbers of leapfrog steps and step sizes using sequential or parallel leapfrog integration (fig. 5B). We simulated a batch of four chains and estimated efficiency using effective sample size (ESS) per second using ArviZ [63]. For relatively larger numbers of leapfrog steps, parallel leapfrog integration achieved substantial efficiency gains over sequential leapfrog (fig. 5B), while for the largest step sizes and fewest leapfrog steps sequential HMC was often more efficient. We also compared peak ESS/s across hyperparameter settings for sequential HMC, HMC with parallel leapfrog, and HMC parallelized across the sequence length (each with four chains). In this setting where the optimal number of leapfrog steps is typically small, parallelizing HMC across the sequence achieved the highest ESS/s. While fewer leapfrog steps were optimal here, the efficiency gains from parallel leapfrog at larger leapfrog step counts may prove valuable in other problems. Appendix C further demonstrates parallel leapfrog integration for HMC on a 501-dimensional item-response model [64].

## 5.4 Scaling Parallel MCMC

**Memory-Efficient quasi-DEER.** The stochastic quasi-DEER algorithm was critical for our MALA BLR experiments. For parallelizing MALA, Figure 6 compares its wall-clock performance against a quasi-DEER implementation that uses automatic differentiation to compute the Jacobian diagonal, as the diagonal was not easily available in closed-form. The memory-efficient stochastic version achieved over $10\times$ faster wall-clock times and remained in memory in multiple additional settings.

**Early-Stopping.** It is well-established that traditional MCMC algorithms inherently output imperfect, approximate samples of the posterior. As such, a reasonable question is whether we must

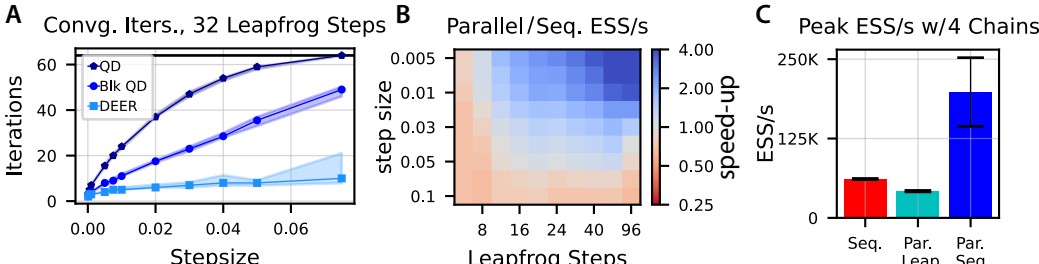

Figure 5: **Parallel HMC sampling targeting a BLR posterior.** **(A)** Block quasi-DEER converges more efficiently than quasi-DEER for parallelizing leapfrog integration. **(B)** Relative ESS/s speedup of parallel vs. sequential leapfrog. **(C)** Peak ESS/s across explored hyperparameter settings for running 4 chains with sequential HMC, HMC with parallel leapfrog, or parallel HMC across the sequence.

run parallel MCMC until full convergence to obtain useful samples for approximating posterior expectations, or if we can get comparable quality samples in much faster time via early-stopping at intermediate Newton iterations. To probe this question, we used parallel MALA as our test case. Indeed, early-stopped parallel MALA can often yield samples with comparable MMD to running parallel MALA until convergence or generating sequential samples (fig. 6B). For example, with $B = 2$ chains, running parallel MALA for only 8 iterations would have yielded samples with near comparable MMD quality to those at convergence, but also with nearly an order of magnitude faster runtime. However, early-stopped parallel MALA performance varies across settings (Appendix C.1) and the optimal number of early-stopping iterations is likely problem-dependent. We leave more thorough investigation of this to future work.

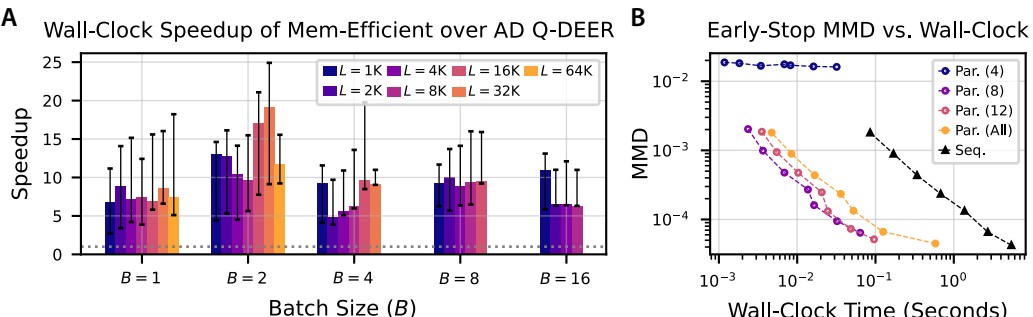

Figure 6: **Effectiveness of methods for scaling parallel MCMC: memory-efficient quasi-DEER and early stopping.** **(A)** Wall-clock speedup multipliers of memory-efficient quasi-DEER over automatic differentiation (AD) quasi-DEER (dotted line is equal runtime). Error bars denote 90% confidence intervals over 20 random seeds. *Memory-efficient quasi-DEER consistently achieves over $10\times$ faster wall-clock times.* AD quasi-DEER ran out of memory for $(B, L) \in \{(4, 64K), (8, 32K), (16, 16K+)\}$, so no bars appear for these cases. **(B)** MMD–wall-clock tradeoffs for $B = 2$ Parallel MALA with/without early stopping. Points on the same line share the same Newton iteration count (4, 8, 12, or full convergence); leftmost and rightmost points correspond to $L = 1K$ and $L = 64K$, respectively. Points closer to the bottom-left indicate greater compute-time efficiency. *One can often achieve comparable MMD with far fewer Newton iterations than needed for full convergence.*

### 5.5 Sentiment Classification from LLM Embeddings

We next scaled parallel MALA to a higher-dimensional sentiment classification task. We encoded reviews from the IMDB dataset [65] into 768-dimensional embeddings using `gemini-embedding-001` [66], partially inspired by Harrison et al. [67]. We targeted a BLR model with a ridge prior that predicted binary sentiment for 1024 randomly selected reviews given the embeddings. We simulated $B = 4$ chains with 4096 samples using sequential and parallel MALA with a step size of $\epsilon = 0.015$. To handle the larger dimensionality, we applied the sliding window method from Section 3.4 (considering window sizes of 128, 256, 512, and 1024) and used a precomputed orthogonal basis transformation from the left singular vectors of the feature covariance matrix. Figure 7A depicts the sliding window approach and shows convergence of parallel samples to the sequential trace. We found that parallel MALA achieved the fastest runtime with a window size of 256 (Figure 7B) and achieved over $3\times$ faster wall-clock time than sequential MALA (Figure 7B-D).

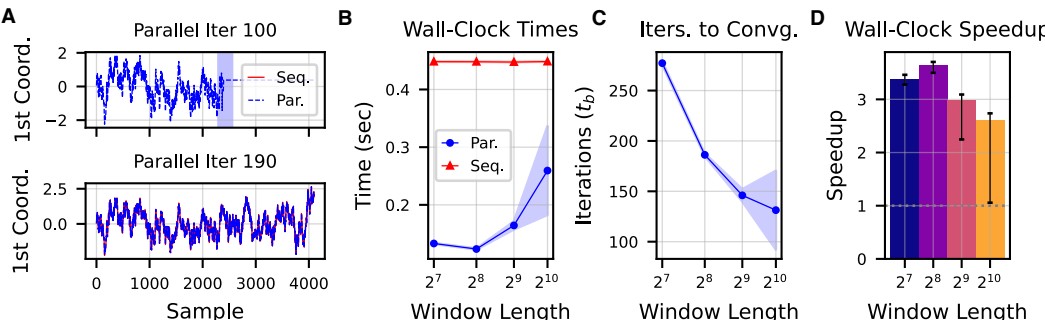

Figure 7: **Sentiment classification from LLM embeddings.** **(A)** Parallel and sequential samples of the first coordinate of the posterior at an intermediate iteration (*top*) and at convergence (*below*). The intermediate iteration shows the sliding window approach: the blue-shaded area is the current window where the trace is updated. **(B)** Wall-clock times for parallel and sequential MALA for sampling 4 chains as a function of window length. **(C)** Number of iterations to convergence for Parallel MALA as a function of window length. **(D)** Wall-clock speedup multiplier of parallel over sequential MALA for different window lengths.

## 6    Conclusion

In this paper, we introduced a novel framework for parallelizing MCMC across the sequence length using parallel Newton iterations. We demonstrated how several widely-used MCMC algorithms can be parallelized using this framework. In multiple experiments, we showed this approach can generate many thousands of samples in tens of iterations, without sacrificing sample quality. In some cases, our approach generated samples an order of magnitude faster than traditional sequential methods. To obtain these results, we developed more efficient approaches for the parallelization of nonlinear recurrences, which is of independent interest. Additionally, this framework provides new opportunities and tradeoffs for the allocation of compute resources. First, one can now tradeoff parallelization of both the batch size and the sequence length. Second, for early-stopping MCMC we provide the ability to dynamically adapt the number of parallel iterations. Ultimately, this approach offers a broadly applicable means of accelerating MCMC, a workhorse of modern statistics.

**Limitations and Future Work.** While our results demonstrate promising acceleration of MCMC via parallel Newton's method, several potential limitations remain. We found that convergence of Newton's method can vary with the sampler step size, target distribution geometry, and Newton-method hyperparameters. Our experiments also primarily targeted unimodal distributions; although we efficiently parallelized sampling of a mixture of Gaussians (Appendix B.5), highly-multimodal targets may have unstable local Jacobians, requiring damped updates or alternative techniques [8]. Future work should establish a theory of when MCMC sampling is efficiently parallelizable and provide guiding principles for adaptive hyperparameter selection. A clear opportunity is to leverage results of contractivity in MCMC [68, 69] and to adapt concurrently-developed theory on DEER convergence [70] to the MCMC setting.

Next, the presented algorithms trade increased memory and computational cost for reduced time complexity by evaluating many additional functions and gradients in parallel. For target distributions with high memory requirements, parallel MCMC may be less advantageous. Moreover, it remains unclear whether allocating parallel resources to more chains or to longer chains yields greater benefit: this is an open question warranting further theoretical and empirical study. In general, future work should scale these methods to higher-dimensional and more challenging targets, and explore specialized hardware-aware acceleration strategies to further improve efficiency.

Our HMC experiments used fixed step sizes and trajectory lengths, unlike adaptive HMC samplers such as NUTS [19]. Because NUTS introduces additional control flow to build sets of candidate points, future work should examine its compatibility with the Jacobians required by DEER. However, simpler dynamic HMC variants that randomly jitter step sizes and leapfrog step counts should remain compatible with our methods. Beyond these HMC extensions, future directions also include applying this framework to stochastic gradient samplers [71–73] and recent MCMC algorithms [74, 75].

Finally, the proposed algorithms have several immediate applications. They can accelerate machine learning systems based on Langevin dynamics [76–78], enable faster approximate inference via early stopping, and improve training algorithms that rely on MCMC sampling, such as Monte Carlo EM and contrastive divergence [79]. In general, we envision this work unlocking MCMC sampling in domains where sequential sampling was previously too time-consuming.

## Acknowledgments and Disclosure of Funding

We thank Art Owen, Kelly Buchanan, and members of the Linderman Lab for helpful feedback. This work was supported by grants from the NIH BRAIN Initiative (U19NS113201, R01NS131987, & RF1MH133778) and the NSF/NIH CRCNS Program (R01NS130789). S.W. would like to acknowledge support from an NSF Graduate Research Fellowship. X.G. would also like to acknowledge support from the Walter Byers Graduate Scholarship from the NCAA. L.K. was a Goldstine Fellow at IBM Research while working on this paper. S.W.L. is supported by fellowships from the Simons Collaboration on the Global Brain, the Alfred P. Sloan Foundation, and the McKnight Foundation. The authors have no competing interests to declare.

Some of the experiments were performed on the Stanford Sherlock computing cluster. We thank Stanford University and the Stanford Research Computing Center for providing computational resources and support that contributed to these research results.

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

# A  Additional Algorithm Details

## A.1  DEER Algorithm

The high-level steps of the DEER and quasi-DEER algorithms are shown in Algorithm 1, where $\star$DEER is either the DEER or quasi-DEER Newton update.

---

**Algorithm 1** Parallel evaluation of nonlinear sequence models using variants of DEER

---

**Require:** initial state $\mathbf{s}_0$, function sequence $f_{1:T}$, initial guess $\mathbf{s}_{1:T}^{(0)}$, tolerance $\delta$, update $\star$DEER

$\quad \Delta_{\max} \leftarrow \infty$

$\quad i \leftarrow 0$

$\quad$**while** $\Delta_{\max} > \delta$ **do**

$\quad\quad \mathbf{s}_{1:T}^{(i+1)} = \star\text{DEER}(\mathbf{s}_{1:T}^{(i)}, f_{1:T}, \mathbf{s}_0)$ $\qquad \triangleright$ global update with $\mathcal{O}(\log T)$ parallel time complexity

$\quad\quad \Delta_{\max} \leftarrow \max_t \left\| \mathbf{s}_t^{(i+1)} - \mathbf{s}_t^{(i)} \right\|_\infty$

$\quad\quad i \leftarrow i + 1$

$\quad$**end while**

$\quad$**return** $\mathbf{s}_{1:T}^{(i)}$

---

## A.2  HMC algorithm with parallel leapfrog

For clarity, the sequential HMC algorithm with parallel leapfrog evaluation is written in full in Algorithm 2. The inner loop of L steps of leapfrog integration is evaluated using parallel Newton's method. It is also straightforward to add a diagonal mass matrix.

---

**Algorithm 2** HMC Step with Parallel Leapfrog

---

$\quad$**function** HMC($p(\mathbf{x}), \mathbf{x}^{(i)}, \epsilon, L$)

$\quad\quad \mathbf{x}_0 \leftarrow \mathbf{x}^{(i)}$

$\quad\quad \mathbf{v}_0 \leftarrow \mathcal{N}(\mathbf{0}, \boldsymbol{I})$ $\qquad\qquad\qquad\qquad \triangleright$ sample momentum

$\quad\quad H_0 \leftarrow \frac{1}{2}\mathbf{v_0}^T\mathbf{v_0} - \log p(\mathbf{x}_0)$ $\qquad\qquad \triangleright$ compute energy

$\quad\quad \mathbf{v}_0 \leftarrow \mathbf{v}_0 + \frac{\epsilon}{2}\nabla_\mathbf{x}\log p(\mathbf{x}_0)$ $\qquad \triangleright$ half-step momentum

$\quad\quad$**for** $t = 1 \rightarrow L$ **do** $\qquad\qquad\qquad \triangleright$ evaluate in parallel

$\quad\quad\quad \mathbf{x}_t \leftarrow \mathbf{x}_{t-1} + \epsilon\,\mathbf{v}_{t-1}$

$\quad\quad\quad \mathbf{v}_t \leftarrow \mathbf{v}_{t-1} + \epsilon\,\nabla_\mathbf{x}\log p(\mathbf{x}_t)$

$\quad\quad$**end for**

$\quad\quad \mathbf{v}_L \leftarrow \mathbf{v}_L - \frac{\epsilon}{2}\nabla_\mathbf{x}\log p(\mathbf{x}_L)$ $\qquad \triangleright$ reverse half-step momentum

$\quad\quad H_L \leftarrow \frac{1}{2}\mathbf{v_L}^T\mathbf{v_L} - \log p(\mathbf{x}_L)$ $\qquad\qquad \triangleright$ compute energy

$\quad\quad U \sim \text{Unif}(0, 1)$

$\quad\quad$**if** $U < \min(1, H_L/H_0)$ **then** $\qquad\qquad \triangleright$ Accept-Reject Step

$\quad\quad\quad$**return** $\mathbf{x}^{(i+1)}$

$\quad\quad$**else**

$\quad\quad\quad$**return** $\mathbf{x}^{(i)}$

$\quad\quad$**end if**

$\quad$**end function**

---

## A.3  Block quasi-DEER for HMC with Parallel Leapfrog

Here we discuss parallelizing leapfrog with block quasi-DEER in additional detail. Recall that for HMC with parallel leapfrog, the state of the function being parallelized is the concatenation of the position and momentum states $\mathbf{s}_t = [\mathbf{x}_t, \mathbf{v}_t]$ with $\mathbf{s}_t \in \mathbb{R}^{2D}$. The Jacobian of the leapfrog step used in HMC with parallel leapfrog integration has a block structure:

$$\boldsymbol{J}(\mathbf{s}_{t-1}) = \frac{\partial f}{\partial \mathbf{s}}(\mathbf{s}_{t-1}) = \begin{bmatrix} \frac{\partial \mathbf{x}_t}{\partial \mathbf{x}_{t-1}} & \frac{\partial \mathbf{x}_t}{\partial \mathbf{v}_{t-1}} \\ \frac{\partial \mathbf{v}_t}{\partial \mathbf{x}_{t-1}} & \frac{\partial \mathbf{v}_t}{\partial \mathbf{v}_{t-1}} \end{bmatrix} \tag{12}$$

$$= \begin{bmatrix} \boldsymbol{I}_D & \epsilon\boldsymbol{I}_D \\ \epsilon\nabla_\mathbf{x}^2\log p(\mathbf{x}_{t-1} + \epsilon\mathbf{v}_{t-1}) & \boldsymbol{I}_D + \epsilon^2\nabla_\mathbf{x}^2\log p(\mathbf{x}_{t-1} + \epsilon\mathbf{v}_{t-1}) \end{bmatrix} \tag{13}$$

where $\boldsymbol{I}_D$ is the identity matrix and $\nabla_{\mathbf{x}}^2 \log p(\mathbf{x}_{t-1} + \epsilon \mathbf{v}_{t-1})$ is the Hessian of $\log p(\mathbf{x})$ evaluated at the new position state $\mathbf{x}_t = \mathbf{x}_{t-1} + \epsilon \mathbf{v}_{t-1}$. We note that here we have derived the Jacobian in the absence of a mass matrix, but it is straightforward to add such a matrix and we use a diagonal mass matrix in the item-response theory experiment. The top two blocks of this Jacobian are diagonal. While the lower blocks depend on the target distribution, the lower-right block is also equal to a diagonal matrix plus the Hessian term scaled by $\epsilon^2$ such that it will be close to diagonal for small step sizes. The structure of this Jacobian motivated the block quasi-DEER approximate Jacobian, which retains only the diagonal of each block

$$\boldsymbol{J}_{BQ}(\mathbf{s}_{t-1}) = \begin{bmatrix} \mathrm{diag}(\frac{\partial \mathbf{x}_t}{\partial \mathbf{x}_{t-1}}) & \mathrm{diag}(\frac{\partial \mathbf{x}_t}{\partial \mathbf{v}_{t-1}}) \\ \mathrm{diag}(\frac{\partial \mathbf{v}_t}{\partial \mathbf{x}_{t-1}}) & \mathrm{diag}(\frac{\partial \mathbf{v}_t}{\partial \mathbf{v}_{t-1}}) \end{bmatrix}. \tag{14}$$

For parallelizing leapfrog integration in HMC, this approximate Jacobian is

$$\boldsymbol{J}_{\mathrm{BQ}}(\mathbf{s}_{t-1}) = \begin{bmatrix} \boldsymbol{I}_D & \epsilon \boldsymbol{I}_D \\ \epsilon \, \mathrm{diag}(\nabla_{\mathbf{x}}^2 \log p(\mathbf{x}_{t-1} + \epsilon \mathbf{v}_{t-1})) & \boldsymbol{I}_D + \epsilon^2 \, \mathrm{diag}(\nabla_{\mathbf{x}}^2 \log p(\mathbf{x}_{t-1} + \epsilon \mathbf{v}_{t-1})) \end{bmatrix}. \tag{15}$$

Importantly, we are efficiently able to construct a single-sample stochastic estimate of the block quasi-DEER Jacobian using only one Hessian-vector product since

$$\mathrm{diag}(\nabla_{\mathbf{x}}^2 \log p(\mathbf{x}_{t-1} + \epsilon \mathbf{v}_{t-1})) = \mathbb{E}_{\mathbf{z} \sim \mathrm{Rad}}[\mathbf{z} \odot (\nabla_{\mathbf{x}}^2 \log p(\mathbf{x}_{t-1} + \epsilon \mathbf{v}_{t-1}) \, \mathbf{z})]. \tag{16}$$

The Hessian-vector product can be efficiently computed using a combination of reverse-mode and forward-mode autodifferentiation.

An important feature of quasi-DEER, which retains only the diagonal of the Jacobian, is it can be implemented using a parallel linear recursion with $\mathcal{O}(TD)$ memory and $\mathcal{O}(TD)$ work. Here we show how block-quasi DEER also admits such a recursion with a constant additional factor of memory and work. Consider two matrices $\boldsymbol{A}$ and $\boldsymbol{B}$ that are each two-by-two block matrices and each block is diagonal with

$$\boldsymbol{A} = \begin{bmatrix} a_1 & & & b_1 & & \\ & \ddots & & & \ddots & \\ & & a_D & & & b_D \\ c_1 & & & d_1 & & \\ & \ddots & & & \ddots & \\ & & c_D & & & d_D \end{bmatrix} = \begin{bmatrix} \mathrm{diag}(\mathbf{a}) & \mathrm{diag}(\mathbf{b}) \\ \mathrm{diag}(\mathbf{c}) & \mathrm{diag}(\mathbf{d}) \end{bmatrix} \tag{17}$$

and

$$\boldsymbol{B} = \begin{bmatrix} \mathrm{diag}(\mathbf{e}) & \mathrm{diag}(\mathbf{f}) \\ \mathrm{diag}(\mathbf{g}) & \mathrm{diag}(\mathbf{h}). \end{bmatrix} \tag{18}$$

The matrix product of $\boldsymbol{A}$ and $\boldsymbol{B}$ is

$$\boldsymbol{AB} = \begin{bmatrix} \mathrm{diag}(\mathbf{a} \odot \mathbf{e} + \mathbf{b} \odot \mathbf{g}) & \mathrm{diag}(\mathbf{a} \odot \mathbf{f} + \mathbf{b} \odot \mathbf{h}) \\ \mathrm{diag}(\mathbf{c} \odot \mathbf{e} + \mathbf{d} \odot \mathbf{g}) & \mathrm{diag}(\mathbf{c} \odot \mathbf{f} + \mathbf{d} \odot \mathbf{h}). \end{bmatrix} \tag{19}$$

Notably, the product of $\boldsymbol{A}$ and $\boldsymbol{B}$ is also a block matrix composed of diagonal blocks and this product can be computed with 8 element-wise products of size $D$. When the matrix $\boldsymbol{A}$ is of size $2D \times 2D$, the linear recursion requires $\mathcal{O}(T4D)$ memory to store the diagonal and two off-diagonals for each time point and $\mathcal{O}(T8D)$ work compared to $\mathcal{O}(T2D)$ memory and $\mathcal{O}(T2D)$ work for a linear recursion of purely diagonal matrices. The resulting linear complexity in $D$ dramatically improves over the cubic complexity in $D$ for DEER when the dimension of the system is large. We hypothesize that this parallel linear recursion used for block quasi-DEER could be useful in other applications of parallelizing nonlinear dynamics with interacting variables and in deep state space models. Additionally, while we constructed block quasi-DEER with a $2 \times 2$ block matrix for this problem, we also note that a similar scheme is possible for block matrices that consist of $k \times k$ blocks with each block being diagonal. We note that this approach has $\mathcal{O}(TkD)$ memory and $\mathcal{O}(Tk^2D)$ work, and so provides a natural interpolation between standard quasi-DEER, with $\mathcal{O}(TD)$ memory and $\mathcal{O}(TD)$ work, and standard DEER, with $\mathcal{O}(TD^2)$ memory and $\mathcal{O}(TD^3)$ work.

# B  Additional Experiment Details

In our applications we use a combination of absolute and relative tolerance [80] as the stopping criterion for parallel Newton's method. In all experiments, we set the absolute tolerance to $10^{-4}$ and relative tolerance to $10^{-3}$ unless otherwise noted.

## B.1  Parallel MALA

We use Bayesian logistic regression (BLR) on the German Credit Dataset (with whitened covariates) [60, 61] as our testbed for this set of experiments. Specifically, we will use quasi-DEER-based Parallel MALA to generate $B$ independent chains of $L$ samples each from the posterior distribution of the BLR model with a $\mathcal{N}(\mathbf{0}, \boldsymbol{I})$ prior on the covariates. We vary the batch size $B \in \{1, 2, 4, 8, 16, 32\}$ and the sequence length $L \in \{1000, 2000, 4000, 8000, 16000, 32000, 64000\}$. To solidify confidence in our results, we will repeat all settings across 20 random seeds. We repeat the corresponding settings for the baseline sequential MALA algorithm. To ensure fairness, we implement the sequential procedure using the JIT-compilable `jax.lax.scan` module.

For both parallel and sequential MALA, we use a learning rate of $\epsilon = 0.0015$, selected to maintain a roughly $\approx 80\%$ acceptance ratio. For both algorithms, we sample independent initial states from their $\mathcal{N}(\mathbf{0}, \boldsymbol{I})$ priors for each of our $B$ chains. Then, we run $w = 3$ burn-in steps to orient these initial states into more statistically-plausible regions of the parameter space. We then set our initial guess of states for all timepoints at Newton iteration 0 to the initial state such that $\mathbf{s}_{1:T}^{(0)} = \mathbf{s}_0$.

For parallel MALA, we set a `max_iter` of $50 + 5L \times 10^{-4}$, chosen heuristically because we expect longer sequence lengths $L$ to require more iterations until all sequence elements numerically converge. We set the absolute tolerance to $5 \times 10^{-4}$. From Appendix Figures 11 and 15, we see that the vast majority of our trials across settings reach numerical convergence before their corresponding `max_iter` values. This suggests that our `max_iter` cutoffs were appropriate.

To compute wall-clock times, for each random seed, we first perform 3 warm-up runs of each evaluated algorithm. Then, we perform 5 timing runs of each evaluated algorithm, logging the total wall-clock time across the 5 timing runs. Finally, we divide this total time by 5 to return our wall-clock time for this seed. The warm-up runs serve to account for potential slowdowns (of both the sequential and parallel MALA) from JIT-compilation overhead on initial executions. We average over 5 timing runs to reduce variance from uncontrollable system processes. We do note, qualitatively, that the one-time cost of JIT-compilation did seem to be noticeably higher for our parallel algorithms compared to the sequential baselines. However, in typical use cases involving repeated sampling (i.e. repeated sampling calls), runtime over many calls will almost certainly dominate compile time. As such, we believe that our timing methodology is well-justified. Nonetheless, reducing this initial overhead remains a useful direction of future work.

For all experiments measuring wall-clock time, we set `full_trace=False` for our quasi-DEER driver so that we only store the output after the last Gauss-Newton iteration. Each run of quasi-DEER under this setup also outputs the numbers of iterations required for convergence of each of our $B$ chains, allowing us to generate Appendix Figure 11. To generate Figures 14 and 15, we require access to the quasi-DEER samples after each Newton iteration, not just after convergence or hitting the maximum iteration count. As such, for these experiments, we set `full_trace=True`.

All experiments were run on an NVIDIA H100 GPU with 80 GB of RAM.

### B.1.1  Measuring Sample Quality via Maximum Mean Discrepancy (MMD)

To assess sample quality, we use Maximum Mean Discrepancy [62]. We do not use traditional MCMC metrics like effective sample size (ESS) or $\hat{R}$ because such metrics mainly target the autocorrelatedness of traditional sequential MCMC algorithms. In contrast, our parallel MALA algorithm by nature does not proceed sequentially sample-by-sample, and the autocorrelation of its generated samples are inherently limited by the autocorrelation of the base sequential MALA algorithm.

Given that the dominant use case for generating MCMC samples is to use said samples towards approximating the posterior expectation of some target function of interest, we can quantify sample quality by how closely our sample-based estimates can match the true target posterior expectations.

Formally, for probability distributions $p_1, p_2$ and function class $\mathcal{F}$, the (theoretical) Maximum Mean Discrepancy (MMD) is defined by Gretton et al. [62] as

$$\text{MMD}(\mathcal{F}, p_1, p_2) = \sup_{f \in \mathcal{F}} \left( \mathbb{E}_{p_1}[f(X)] - \mathbb{E}_{p_2}[f(Y)] \right),$$

for $X \sim p_1$ and $Y \sim p_2$. If we set $p_2$ to be the true target posterior distribution and let $p_1$ represent the distribution represented by our generated samples, then MMD is a natural measure of sample quality.

However, we do not have access to the true target posterior distribution $p_2$. Thus, we approximate the "ground truth" $p_2$ via a large set of *gold standard* MCMC samples generated using the state-of-the-art No U-Turn Sampler with Dual-Averaging Stepsize Adaptation introduced by [19] and implemented in `TensorFlow Probability` [81]. Specifically, following standard best practices, we aim for a target acceptance probability of $75\%$ with initial states being drawn from the $\mathcal{N}(\mathbf{0}, \boldsymbol{I})$ prior. To ensure high quality, we perform $50{,}000$ burn-in steps before collecting $100{,}000$ samples as our *gold standard* approximation of the true target posterior.

Yet, taking the supremum over all $f$ in a potentially-uncountable function class $\mathcal{F}$ is analytically intractable. As such, following Gretton et al. [62], we pick a function class $\mathcal{F}$ corresponding to a reproducing kernel — in our case, we will use a Gaussian RBF kernel:

$$k(\boldsymbol{x}_1, \boldsymbol{x}_2) := \exp\left( -\frac{\|\boldsymbol{x}_1 - \boldsymbol{x}_2\|_2^2}{2\sigma^2} \right),$$

with hyperparameter $\sigma$ to be discussed in the next paragraph. Let $\{\boldsymbol{x}_i\}_{i=1}^N$ refer to the samples outputted by sequential or parallel MALA, with $N = B \times L$ and let $\{\boldsymbol{y}_j\}_{j=1}^M$ represent the *gold standard* samples. In such a setting, we can empirically and unbiasedly estimate the MMD of our samples with respect to the true target posterior as follows (see [62]):

$$\widehat{\text{MMD}}(\mathcal{F}_k, p_1, p_2) = \frac{1}{N(N-1)} \sum_{i=1}^N \sum_{j \neq i}^N k(\boldsymbol{x}_i, \boldsymbol{x}_j)$$

$$+ \frac{1}{M(M-1)} \sum_{i=1}^M \sum_{j \neq i}^M k(\boldsymbol{y}_i, \boldsymbol{y}_j) - \frac{2}{MN} \sum_{i=1}^M \sum_{j=1}^N k(\boldsymbol{x}_i, \boldsymbol{y}_j).$$

Because we are working with high-dimensional settings with large batch sizes $B$ and sequence length $L$, we cannot store all of our samples in GPU memory at once. As such, we approximate each term above using at most $M_{\max} = 50{,}000$ randomly-selected samples from our MALA outputs and take the average of our $\widehat{\text{MMD}}$ estimates over 10 replications.

Finally, following Gretton et al. [62], we select $\sigma$ to be the median pairwise distance between samples in our *gold standard* set. Because storing the entire pairwise distance matrix would overwhelm our GPU memory, we randomly select $50{,}000$ samples from this set, compute the median pairwise distance of this subset, and then take the mean over $10{,}000$ repetitions of this procedure.

## B.2 Gibbs sampling experiment

The generative model for the Gibbs sampling experiment is

$$\tau^2 \sim \chi^{-2}(\nu_0, \tau_0^2) \qquad\qquad \mu \sim \mathcal{N}(\mu_0, \tau^2/\kappa_0) \tag{20}$$

$$\theta_s \sim \mathcal{N}(\mu, \tau^2) \qquad\qquad \sigma_s^2 \sim \chi^{-2}(\alpha_0, \sigma_0^2) \qquad\qquad x_{s,n} \sim \mathcal{N}(\theta_s, \sigma_s^2) \tag{21}$$

where $\nu_0 = 0.1$, $\tau_0^2 = 100.0$, $\mu_0 = 0.0$, $\kappa_0 = 0.1$, $\alpha_0 = 0.1$, and $\sigma_0^2 = 10.0$. We parallelized Gibbs sampling coordinate updates that simulate samples from the 18-dimensional posterior $p(\tau^2, \mu, \{\theta_s, \sigma_s^2\}_{s=1}^S \mid \mathbf{X}, \phi)$ where $\mathbf{X}$ are the set of observations across the $S = 8$ schools each with $N = 20$ students and the hyperparameters are $\phi = \{\nu_0, \tau_0, \mu_0, \kappa_0, \alpha_0, \sigma_0^2\}$. We simulated data such that the means and standard errors corresponded to those in Table 5.2 of Gelman and Hill [64]. The dynamics function at time-step $t$ proceeded via the following coordinate updates

$$\tau_t^2 \sim p(\tau_t^2 \mid \mu_{t-1}, \{\theta_{s,t-1}\}_{s=1}^S, \phi) \tag{22}$$

$$\mu_t \sim p(\mu_t \mid \{\theta_{s,t-1}\}_{s=1}^S, \tau_t^2, \phi) \tag{23}$$

$$\theta_{s,t} \sim p(\theta_{s,t} \mid \mu_t, \tau_t^2, \sigma_{s,t-1}^2, \mathbf{X}, \phi) \tag{24}$$

$$\sigma_{s,t}^2 \sim p(\sigma_{s,t}^2 \mid \theta_{s,t}, \mathbf{X}, \phi) \tag{25}$$

where $\theta_{s,t}$ denotes the sampled value for $\theta_s$ at time point $t$. We reparameterized this update using standard reparameterization for Gaussian random variables and the fact that if $z \sim \chi^{-2}(\nu, \tau^2)$ then $kz \sim \chi^{-2}(\nu, k\tau^2)$ for the $\tau^2$ and $\sigma^2$ given that the location parameter of the $\chi^{-2}$ distributions were fixed and known for each of those variables. In our implementation, we passed a random number key as input to each step of the function and sampled random variates as part of $f_t$.

For each chain, we first simulated a draw from the prior distribution and then ran one sequential sampling step targeting the posterior. The result of this sequential sampling step was set to be the initial state $\mathbf{s}_0$ for each chain. For Newton's method, the guess of the state sequence at iteration $0$ was set to the initial state such that $\mathbf{s}_t^{(0)} = \mathbf{s}_0$ for all $t = 1, \ldots, T$, as in the MALA experiment. When using quasi-DEER for this problem, we used a 3-sample stochastic estimate of the diagonal. We additionally applied a tuned diagonal preconditioner to the stochastic quasi-DEER Jacobian estimates.

To compare the timing of sequential and parallel sampling, we ran chains of batch size $B \in \{1, 2, 4, 8, 16, 32, 64\}$ and length $L \in \{8000, 16000, 32000, 64000, 128000\}$. For all batch sizes except for $B = 64$, we additionally simulated chains of length $L \in \{256000, 512000, 1024000\}$. On a CPU, we ran sequential sampling for all batch sizes and for lengths up to $L = 32000$; for values of $L > 32000$ we extrapolated the time for sequential sampling (i.e. for $L = 64000$ we doubled the amount of time it took to simulate $L = 32000$ samples). We repeated each run 5 times and computed the timing in a similar manner as our MALA experiments. That is, we computed the average time to run the JIT-compiled code across 3 timing runs after performing 2 warm-up runs of each algorithm.

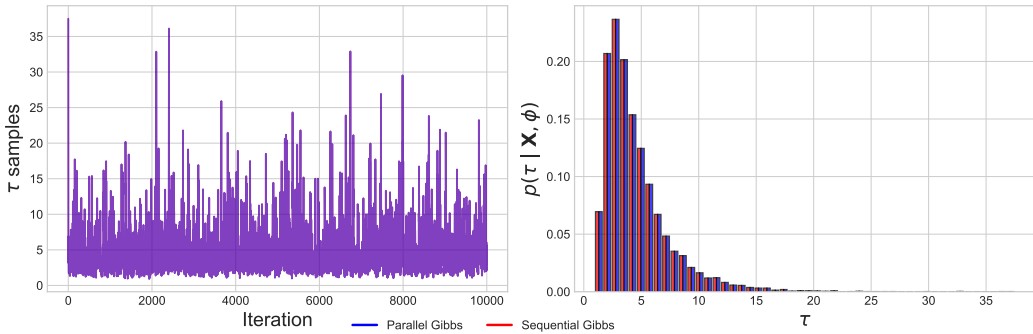

Figure 8: **Example samples from sequential and parallel Gibbs sampling.** *Left:* Samples of $\tau$ across iterations for sequential and parallel Gibbs sampling. The two sample traces are overlapping. We note that we transformed $\tau^2$ into $\tau$ for visualization purposes. *Right:* The corresponding histograms of samples of $p(\tau \mid \mathbf{X}, \phi)$ are identical.

### B.3 Sequential HMC with parallel leapfrog experimental details

We also applied sequential HMC with parallel leapfrog to the Bayesian logistic regression model of the German-Credit dataset [60, 61] with whitened covariates. We focused on sampling 4 batched chains and swept across the number of leapfrog steps $L = \{4, 8, 12, 16, 20, 24, 32, 40, 64, 96\}$ and step sizes $\epsilon = \{0.005, 0.0075, 0.01, 0.02, 0.03, 0.04, 0.05, 0.075, 0.1\}$. Step sizes at $0.1$ and above resulted in unstable integration and acceptance rates near zero. For each combination of $L$ and $\epsilon$, we estimated the time to simulate 2000 samples from HMC using either sequential or parallel leapfrog integration using a JIT-compiled function. We repeated each step size and number of leapfrog steps across 10 random seeds. In all cases, we only used the absolute tolerance. For each run, we computed the average effective sample size across parameters. We then reported the average effective sample size per second. A relative comparison of the average ESS/s across runs between HMC with parallel or sequential leapfrog is reported in Figure 5B. Additionally, the maximum ESS/s across settings of $L$ and $\epsilon$ for HMC with parallel or sequential leapfrog is shown in Figure 5C.

When parallelizing the leapfrog algorithm, the initial state is the concatenation of the initial position and the momentum after a half-step $\mathbf{s}_0 = [\mathbf{x}_0, \mathbf{v}_0]$. We set the guess of the sequence of states to be equal to this initial state, as in the Gibbs and MALA setups. Across 10 different initial states, compared the number of iterations until convergence for DEER, block quasi-DEER, and quasi-DEER

across the same set of step sizes and with $L = 32$ steps for BLR. These results are reported in Figure 5A.

We next explored using sequential HMC with parallel leapfrog for an item-repsonse theory model [64, 61]. The target distribution in this case has $D = 501$ dimensions and the dataset has approximately 30K datapoints, making this a substantially larger problem. We compared the time to run HMC with either sequential or parallel leapfrog across 1 or 2 chains and varying step sizes and number of leapfrog steps. For each setting we ran 10 repeats across different random number seeds. We estimated the time to simulate 1000 samples from the posterior distribution (Figure 19). Here, we used a pre-tuned diagonal mass matrix, which empirically we found helped improve the convergence of parallel leapfrog integration. We found that when simulating 1 chain the HMC with parallel leapfrog can achieve speed-ups over HMC with sequential leapfrog. However, at two chains the sequential leapfrog was faster.

### B.3.1 Step-size for HMC with parallel leapfrog

Tuning the step size and number of leapfrog in HMC is important for optimal exploration of the target distribution. The standard analyses for determining the optimal step size and acceptance rate typically assume a linear time and computational cost in the number of leapfrog steps [12, 82]. However, parallelizing the leapfrog integration can enable sublinear cost in the number of steps, violating this assumption. A second compounding factor is that the number of parallel Newton iterations until convergence of the parallel integration can vary as a function of the step size, as we saw in our results. New analysis investigating the appropriate optimal step sizes and acceptance rates need to be devised under these conditions. We hypothesize that in some cases, these conditions will favor the use of smaller step sizes and more leapfrog steps, thereby increasing the optimal acceptance rate. We currently leave a formal analysis of this to future work.

### B.4 Parallel HMC across the sequence

We parallelized HMC across the sequence for both the Bayesian logistic regression example and for the Rosenbrock distribution. For parallel HMC on Bayesian logistic regression we used memory-efficient quasi-DEER to parallelize the algorithm. We tuned the step size, number of leapfrog steps, and hyperparameters of the algorithm to consistently achieve fast convergence with high numbers of samples. The resulting parameters were $L = 4$ leapfrog steps and step size $\epsilon = 0.04$. We simulated five sequential samples before setting the initial state for parallel HMC. In this case, we simulated batches of 4 chains of length 1000. The initial guess of the states for parallel Newton's method was set to the initial state for all time points. When using memory-efficient quasi-DEER, we clipped the magnitude of the diagonal Jacobian to be less than 1.0 and we used a damping factor of 0.4 [8]. We repeated the sampler across 10 different seeds. The sampler typically converged in around 20 iterations and the acceptance ratio was around $0.89\%$. We computed the ESS/s in the same manner as for sequential HMC with sequential or parallel leapfrog in the previous subsection.

In Figure 1, we parallelized HMC sampling of the Rosenbrock "banana" distribution implemented in TFDS [61]. We parallelized a chain of 100K samples where each HMC consisted of $L = 8$ leapfrog steps with step size $\epsilon = 0.5$. This resulted in an acceptance rate of approximately $97.6\%$. For this 2D distribution, we used the full DEER Jacobian scaled by a damping factor of 0.5 [8] and clipped to be within $[-1, 1]$. For this motivating example, we ran this with a single random seed and did not examine timing or convergence profiles across runs.

### B.5 Multimodal mixture of Gaussians example

Here we demonstrate that the approach can efficiently parallelize sampling from a multimodal distribution. We constructed a mixture-of-Gaussians distribution in two dimensions with four components (Figure 9). We used MALA with a step size of $\epsilon = 0.1$ to simulate 100K samples from this distribution. Parallel MALA using quasi-DEER with damping (called quasi-ELK in Gonzalez et al. [8]) converged to the ground truth sequential trace in 50 parallel iterations. The resulting set of samples at parallel iteration 50 are shown in Figure 9. While here we visualized the results for quasi ELK, we note that quasi-DEER with clipped diagonal Jacobians between $-1$ and 1 also efficiently converged in 72 parallel iterations. This preliminary result demonstrates the approaches can be used to traverse multimodal distributions.

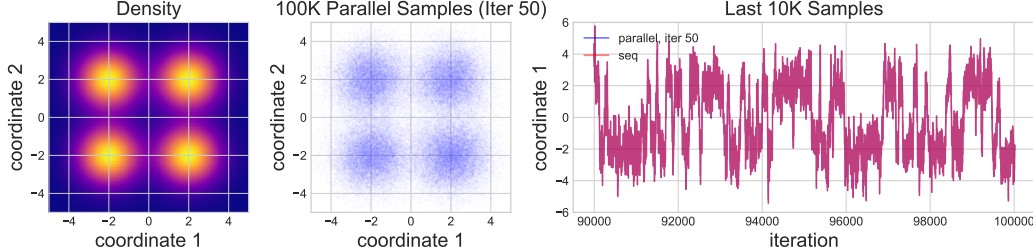

Figure 9: **Parallel sampling of a multimodal mixture of Gaussians distribution.** The left two panels show the target multimodal density and the final samples from running 50 parallel MALA iterations to simulate 100K samples. The rightmost panel shows the last 10K samples from sequential MALA sampling and parallel MALA sampling. Parallel MALA has converged to the ground truth sequential trace of 100K samples in 50 parallel iterations such that the parallel and sequential traces are completely overlapping.

## C    Additional Results Figures

### C.1    Parallel MALA

In this section, we include the following additional results figures for parallel MALA. The intent of these figures is to show trends over full parameter sweeps of batch size $B$ and sequence length $L$ that corroborate and augment our findings in the main text:

- Figure 10: Wall-clock times (in seconds) for parallel MALA vs. sequential MALA sampling to generate $B$ batches of $L$ samples apiece. *This contains the full results corresponding to Panel (A) in Figure 3 of the main text.*

- Figure 11: Distributions of iterations needed for each of $B$ chains to numerically converge in parallel MALA. *This contains the full results corresponding to Panel (B) in Figure 3 of the main text.*

- Figure 12: MMD vs. wall-clock time performance profiles for parallel MALA vs. sequential MALA sampling. *This contains the full results corresponding to Figure 4 of the main text.*

- Figure 13: MMD vs. wall-clock time performance profiles for parallel MALA vs. sequential MALA sampling, interpolating over batchsize $B$. *This figure provides an alternate perspective (grouping by sequence length $L$ instead of batch size $B$) of the results in Figure 12 above and Figure 4 of the main text.*

- Figure 14: Early-stopping MMD vs. wall-clock time tradeoffs. *This figure contains the full results corresponding to Panel (B) in Figure 6 of the main text.*

- Figure 15: Maximum absolute errors accrued by parallel MALA w.r.t. the sequential sampling MALA algorithm as a function of Newton iteration. *The purpose of this figure is to demonstrate that parallel MALA achieves numerical convergence on most combinations of batch size $B$ and sequence length $L$ within the maximum number of iterations.*

- Figure 16: Pooled distributions of iterations needed for parallel MALA chains to converge. *This figure further emphasizes that the vast majority of length-$L$ chains achieve numerical convergence well before exhausting the maximum number of allowed iterations.*

- Figure 17: Example samples of Bayesian logistic regression coefficient $\beta_1$ on a numerically-converged parallel MALA chain. *This figure demonstrates that at numerical-convergence, parallel and sequential MALA indeed output the same samples.*

- Figure 18: Example samples of Bayesian logistic regression coefficient $\beta_1$ on a *not* numerically-converged parallel MALA chain. *The purpose of this figure is to demonstrate that even without complete numerical convergence, parallel MALA's generated samples are still effectively identical to those of sequential MALA's.*

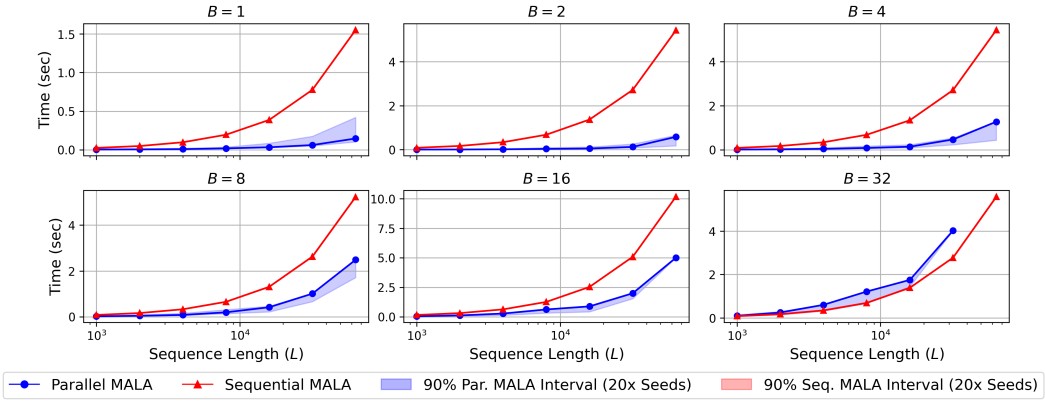

Figure 10: **Wall-clock times (in seconds) for parallel MALA vs. sequential MALA sampling to generate $B$ batches of $L$ samples apiece.** The lightly-colored bands represent 90% confidence intervals over 20x random seeds. For all results shown above, parallel MALA was run until full numerical convergence. This figure contains the full results corresponding to Panel (A) in Figure 3 of the main text. *The main takeaway is that for all sequence lengths $L$ and all batch sizes $B$ except for $B = 32$, parallel MALA is significantly faster than sequential MALA at generating samples.*

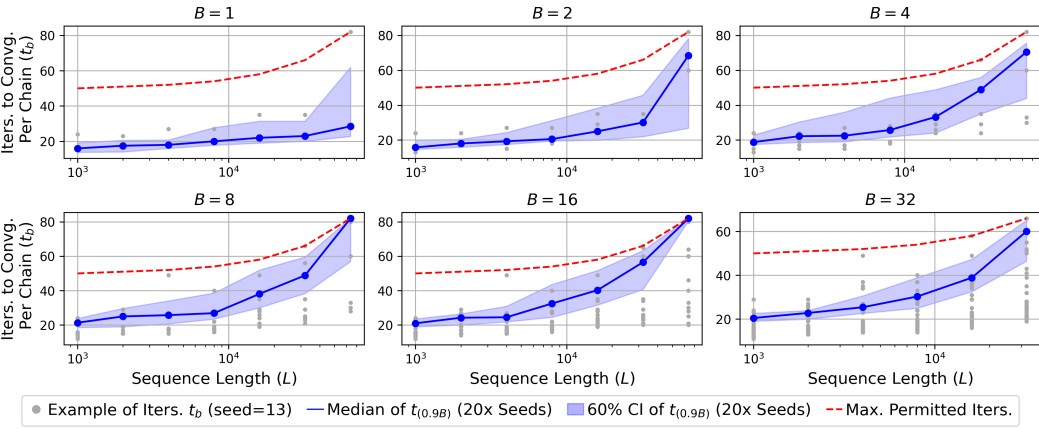

Figure 11: **Distributions of iterations needed for each of $B$ chains to numerically converge in parallel MALA.** The red dotted line represents the maximum number of iterations permitted for a given setting. For a given collection of $B$ chains, let $t_b$ represent the number of iterations needed for chain $b$ of $B$ to converge. Letting $t_{(1)} \leq \cdots \leq t_{(B)}$ represent these numbers of iterations in increasing order, the solid blue line represents the median of $t_{(0.9B)}$ (i.e., the 90th percentile within these $B$ chains) taken across 20x random seeds. The light-blue band represents a 60% confidence interval of $t_{(0.9B)}$ computed across the 20x random seeds. The gray dots represent the observed numbers of iterations until convergence for each chain $b$ in $B$ for one representative seeded trial. This figure contains the full results corresponding to Panel (B) in Figure 3 of the main text. *The main takeaway is that in most settings, parallel MALA converges in only a few dozen iterations despite generating tens, if not hundreds, of thousands of samples. Moreover, in the vast majority of settings, parallel MALA numerically converges before reaching the* `max_iter` *limit.*

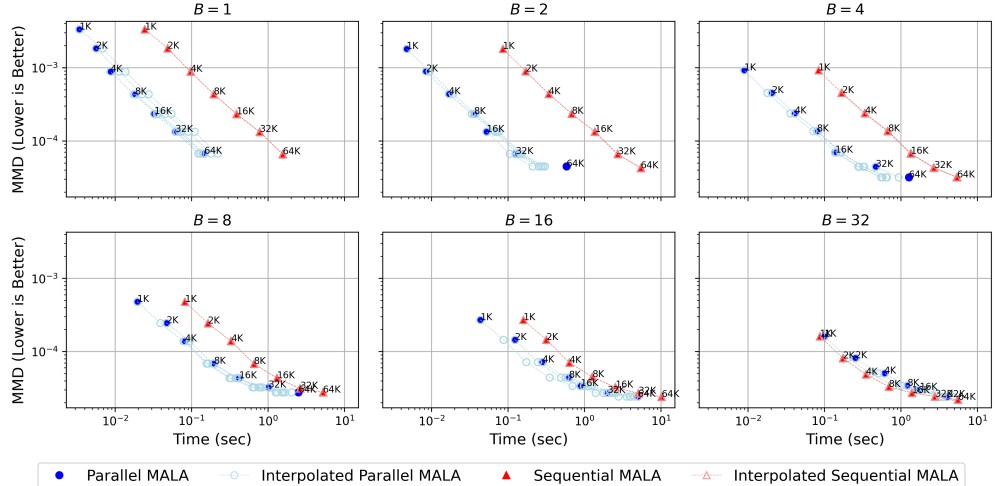

Figure 12: **MMD vs. wall-clock time performance profiles for parallel MALA vs. sequential MALA sampling.** *The purpose of MMD is to measure sample quality. The main takeaway is that for all batch sizes except for $B = 32$, we see that parallel MALA achieves virtually-identical MMD (i.e., outputs equally high quality samples) as sequential sampling, oftentimes an order of magnitude or more faster.* On $B = 32$, we achieve rough parity. The solid blue triangles and solid red circles represent the observed performances of parallel MALA and sequential MALA sampling, respectively, with corresponding sequence lengths $L$ annotated next to each point. In each panel, we also investigate the performance tradeoffs of using smaller sequence lengths, but over multiple serial runs. For example, assuming parallel MALA is run until numerical convergence, running parallel MALA with $B = 4$ and $L = 32K$ should output equivalent sample size and sample quality as running eight instances of parallel MALA with $B = 4$ and $L = 4K$ one after another, using the last samples from each serial run as the initialization for the next instance. Of course running eight instances of $B = 4$ and $L = 4K$ should take eight times longer than running one instance of the same settings. However, using smaller sequence lengths could lead to proportionately faster numerical convergence per instance, making it potentially more time-efficient in certain cases than running one single instance with a longer sequence length. As such, we linearly interpolate to investigate such performance tradeoffs, with the interpolated values shown in the hollow triangles and circles. **For all panels, points closer to the bottom-left corner indicate better compute-time efficiency (i.e, faster sampling and higher quality of samples).** This figure contains the full results corresponding to Figure 4 of the main text, organized by batch size $B$.

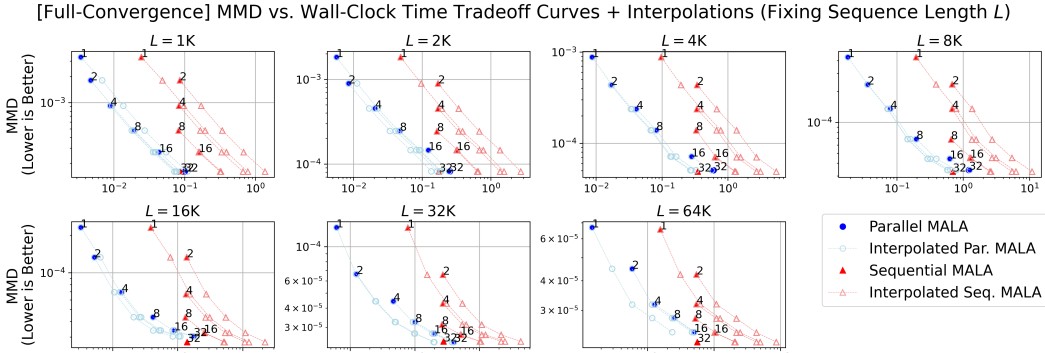

Figure 13: **MMD vs. wall-clock time performance profiles for parallel MALA vs. sequential MALA sampling, interpolating over batch size** $B$**.** Why don't we always use $B = 32$ (or the largest batch size possible)? Holding sequence length $L$ fixed in each subplot, we explore MMD vs. wall-clock-time tradeoffs of using various batch sizes. For example, assuming parallel MALA is run until numerical convergence, running sixteen independent parallel MALA instances of $B = 2$ and $L = 32K$ one after another should yield equivalent sample size and sample quality as running one parallel MALA instance of $B = 32$ and $L = 32K$, and take 16x longer than running one parallel MALA instance of $B = 2$ and $L = 32K$. However, directly running $B = 32$ and $L = 32K$ could overwhelm GPU memory, leading to significantly slower wall-clock performance than serially running smaller batches. This reasoning justifies our interpolation-over-$B$ analyses shown in the hollow triangles and circles. *The interpolations in the $L = 32K$ subplot reveal that running sixteen independent instances of parallel MALA at $L = 32K$ iterations apiece would have achieved equivalent MMD as running parallel MALA or sequential sampling directly at $B = 32$ and $L = 32K$, but with noticeably faster speed.* **For all panels, points closer to the bottom-left corner indicate better compute-time efficiency (i.e, faster sampling and higher quality of samples).** This figure provides an alternate perspective (grouping by sequence length $L$ instead of batch size $B$) of the results in Figure 12 above and Figure 4 of the main text.

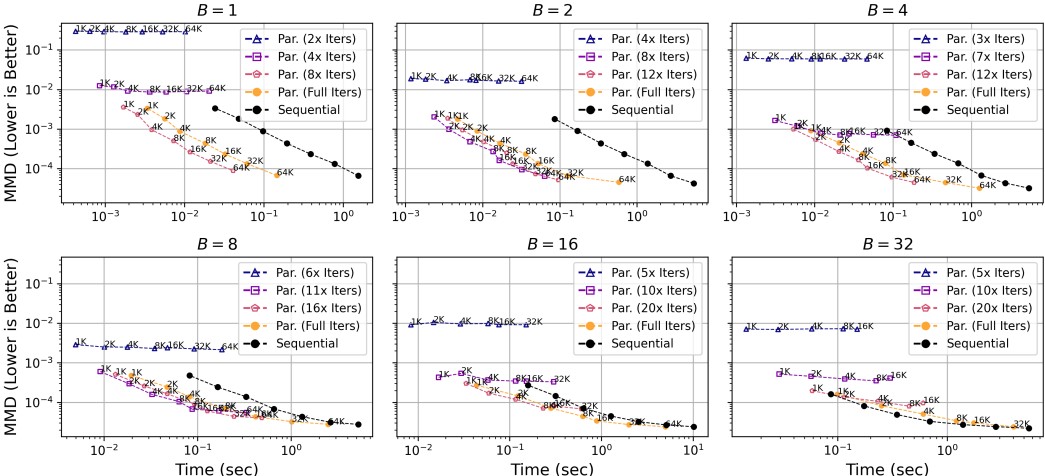

Figure 14: **Early-stopping MMD vs. wall-clock time tradeoffs.** Empirically, we know that the MMD of a batch of parallel MALA sample chains can converge to that sequential MALA's in significantly fewer iterations than are required for the numerical convergence of individual chains. Each color-coded point represents the median MMD and interpolated wall-clock time (across 20x random seeds) of parallel MALA for a given batch size $B$ and sequence length $L$ after running for only a few iterations (see individual subplot legends) vs. running until full numerical convergence ("Full Iters") and the sequential sampling baseline. *The main takeaway is that in many cases, by performing early stopping (i.e., only running for very few parallel MALA iterations instead of waiting until full numerical convergence) one can obtain MCMC samples with comparable MMD sample quality but in much faster time.* This figure contains the full results corresponding to Panel (B) in Figure 6 of the main text. **For all panels, points closer to the bottom-left corner indicate better compute-time efficiency (i.e, faster sampling and higher quality of samples).**

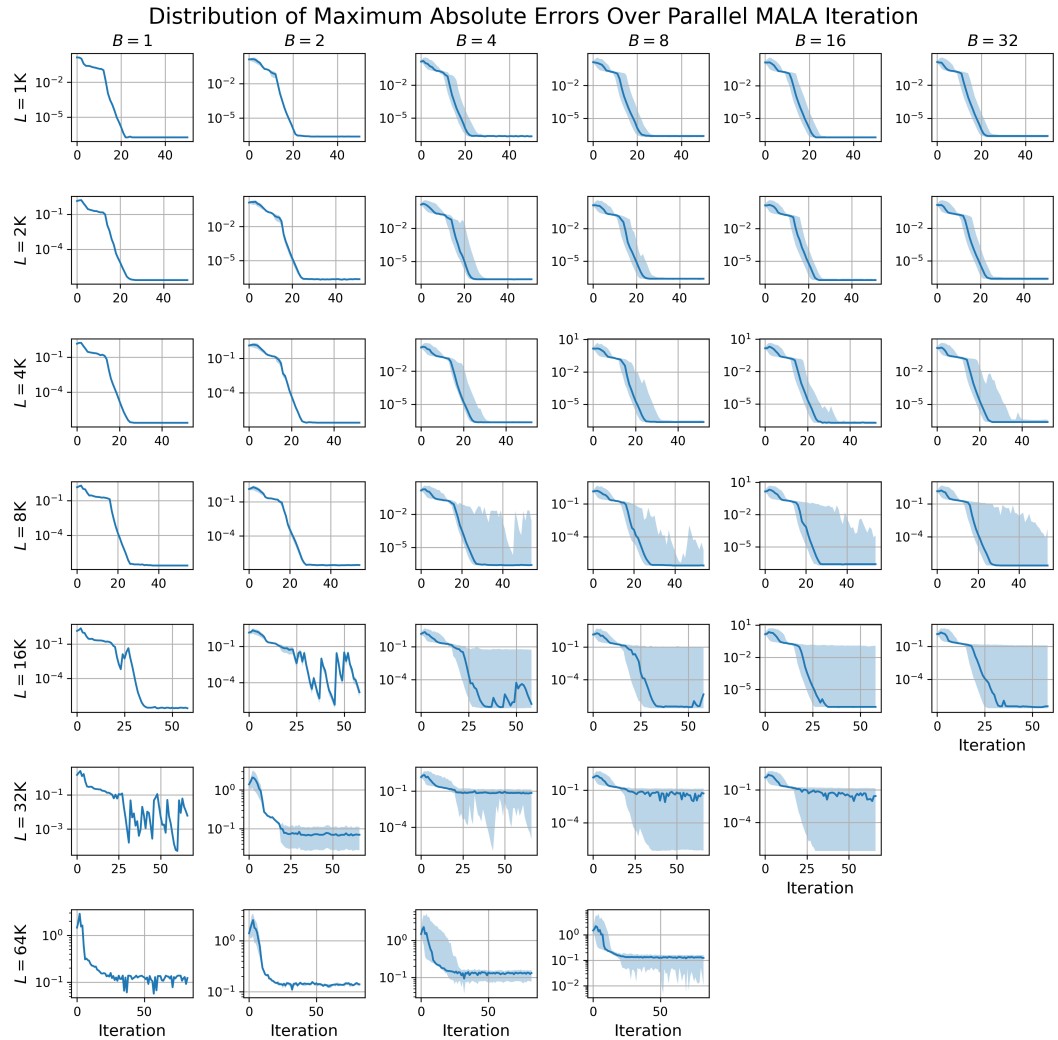

Figure 15: **Maximum absolute errors accrued by parallel MALA w.r.t. the sequential sampling MALA algorithm as a function of Newton iteration.** The solid blue line represents the median of the median maximum absolute error (with the maximum taken over the entire sequence length $L$ and dimensions $D$), with the inner median computed over all chains $b$ in $B$, and the outer median computed over 20x random seeds. The lower and upper limits of the light blue shaded region represent the medians (across 20x random seeds) of the 20th and 80th quantiles of the maximum absolute error (taken over the $B$ chains per seed). *The main takeaway of this figure is that parallel MALA achieves numerical convergence on most combinations of batch size $B$ and sequence length $L$ within the maximum number of iterations.*

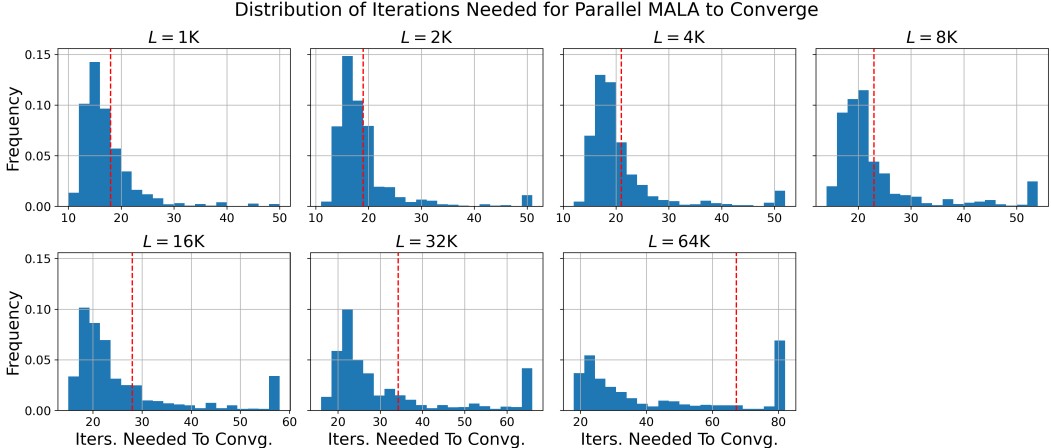

Figure 16: **Pooled distributions of iterations needed for parallel MALA chains to converge.** Each histogram shows the number of iterations required for convergence for all chains with sequence length $L$, pooled across all batch sizes $B$ and 20 random seeds. The red dotted line marks the 75th percentile, and the `max_iter` for each setting of $L$ is marked by the right-most $x$-axis value. This figure emphasizes that the vast majority of length-$L$ chains achieve numerical convergence well before exhausting the maximum number of allowed iterations.

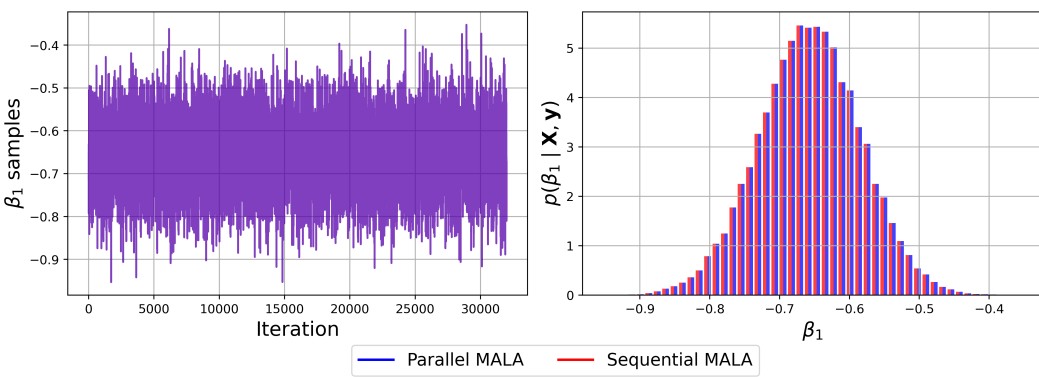

Figure 17: **Example samples of Bayesian logistic regression coefficient $\beta_1$ on a numerically-converged parallel MALA chain.** *Left:* trace plots of $\beta_1$ across iterations for sequential and parallel MALA sampling. The traces complete overlap, confirming that we indeed have numerical convergence. *Right:* the histograms representing the posterior distributions implied by parallel and sequential MALA's samples are identical.

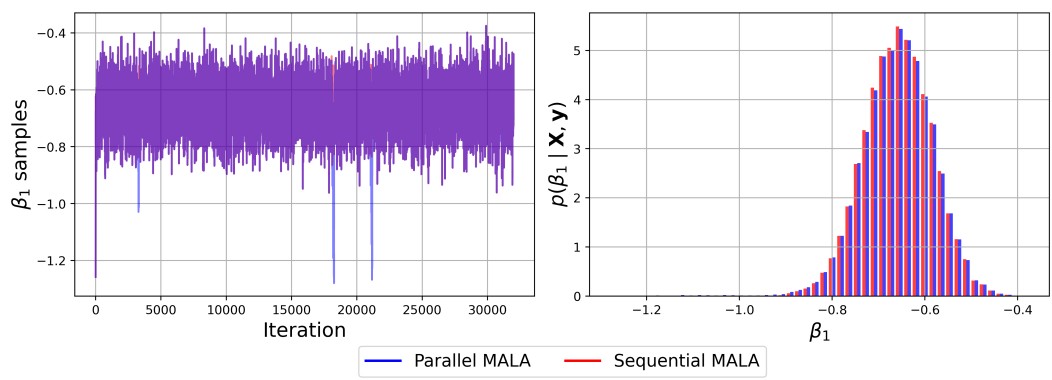

Figure 18: **Example samples of Bayesian logistic regression coefficient** $\beta_1$ **on a *not* numerically-converged parallel MALA chain.** Specifically, this chain reached `max_iters` without numerical convergence. *Left:* trace plots of $\beta_1$ across iterations for sequential and parallel MALA sampling. The traces still effectively completely overlap, implying that the two sets of samples are basically identical. The discrepancy can be seen around iteration 18000. *Right:* despite the lack of numerical convergence, the histograms representing the posterior distributions implied by parallel and sequential MALA's samples are effectively identical. *The main takeaway of this figure is that even without complete numerical convergence, parallel MALA's generated samples are still effectively identical to those of sequential MALA's.*

### C.2 Parallel Leapfrog Integration for HMC

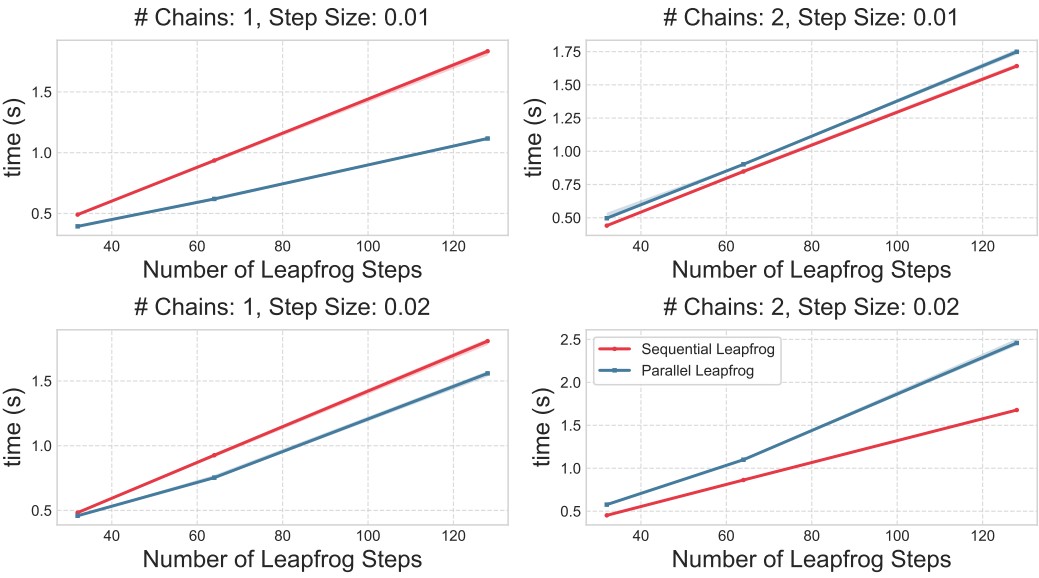

Figure 19: **Time comparison for sequential HMC on the item-response model with sequential or parallel leapfrog integration.** The time to simulate 1000 samples sequentially using HMC with either sequential or parallel leapfrog on the item-response model, as a function of the step size, number of leapfrog steps, and number of chains (panels). The target dimensionality is $D = 501$ for this problem and each likelihood evaluation is computed across approximately 30K datapoints. For one chain, HMC with parallel leapfrog is fastest across the two step sizes. However at two chains we do not see speed-ups when using sequential HMC with parallel leapfrog for this problem.

