# OpenReview forum: "Parallelizing MCMC Across the Sequence Length"
_NeurIPS.cc/2025/Conference — NeurIPS 2025 poster_

### Official Review · Reviewer_hixK · 2025-06-29

**Clarity:** 3
**Significance:** 2
**Originality:** 2
**Rating:** 4
**Confidence:** 4

**Summary:**

This paper tackles the problem of speeding up Markov chain Monte Carlo (MCMC) by parallelizing across the sequential steps of each chain. The authors adapt the DEER algorithm (a parallel Newton-based method originally used for RNN-style recurrences) to solve an entire chain’s updates in parallel. This approach is novel (up to my best knowledge): by performing $O(\log T)$ parallel Newton iterations instead of $T$ sequential steps, they show it’s possible to generate long trajectories with only a few dozen iterations. The key innovation is recasting common MCMC samplers (Gibbs, MALA, HMC) as fixed-point problems that can be solved using parallel Newton-type methods.

The authors develop several algorithms within the DEER framework:
- **DEER**: Uses full Jacobians for quadratic convergence with O(log T) parallel time complexity
- **Quasi-DEER**: A cheap variant using only Jacobian diagonals
- **Stochastic Quasi-DEER**:  reduces the memory footprint by estimating Jacobian diagonals
- **Block Quasi-DEER**: Optimized for HMC's leapfrog integrator structure

The methods demonstrate  10×–30× speedups over sequential sampling across various applications, including hierarchical Gaussian models, Bayesian logistic regression, and complex targets.

This breakthrough transforms MCMC from a fundamentally sequential O(T) process into a parallelizable O(log T) problem, opening new possibilities for scaling Bayesian inference on modern accelerators. The work establishes both theoretical foundations and practical algorithms for leveraging massive parallelism in Monte Carlo sampling.

**Questions:**

-  The paper references the DEER algorithm (from Lim et al. [5]) as part of the parallelization approach, but its presentation is quite limited. Even if DEER is not the central theme, a clearer self-contained explanation of what DEER does and how it integrates into your method would greatly improve readability. For instance, please elaborate on the intuition and steps of the DEER algorithm (and quasi-DEER) in the main text or appendix. This helps readers who are unfamiliar with prior work [5,6] understand the foundation of your parallel MCMC approach without extensive external reading.
- The paper makes several claims (e.g. about convergence guarantees, sublinear scaling, “orders of magnitude” speedups, etc.) that would benefit from stronger theoretical backing or at least references to supporting theory. I encourage the authors to support these claims with background results or analysis. For example, if you claim that the stochastic quasi-DEER “enjoys provable convergence guarantees” or that Newton’s method yields quadratic convergence, please either cite the precise theorems/propositions (from prior work or your appendix) or outline the conditions under which these claims hold.
- While I find the application of parallel Newton/DEER methods to MCMC to be novel and exciting, the manuscript could do more to explicitly position this contribution relative to prior work. I suggest the authors clearly highlight what is new in this work (e.g. parallelizing specific MCMC samplers across chain length, the new quasi-Newton variants, etc.) versus what is adopted from existing methods. For instance, discuss how your approach differs from or improves upon prior parallel MCMC efforts or parallel diffusion simulations [18–22,28,29] mentioned in related work.

**Ethical Concerns:**

["NO or VERY MINOR ethics concerns only"]

**Limitations:**

Yes

**Quality:**

3

**Strengths And Weaknesses:**

Strengths:
- The paper builds on a existing framework (treating the chain update as a fixed-point problem solved by Newton’s method) and extends it with appropriate modifications. For example, the authors acknowledge that the straightforward DEER algorithm would require storing and multiplying through $T$ Jacobian matrices of size $D\times D$, leading to cubic time and quadratic memory in state dimension $D$. They address this by using quasi-Newton updates that only use the diagonal of each Jacobian (or Hutchinson’s stochastic diagonal estimate), reducing complexity to $O(D)$ and making the method much more scalable. They also introduce a block-diagonal Newton scheme for HMC’s leapfrog integrator, which accounts for position–momentum coupling and roughly halves the number of iterations needed to converge compared to a naive diagonal approach.
-  the authors identified potential bottlenecks (memory use, slow convergence) and devised solutions grounded in numerical methods. The approach for handling non-differentiable Metropolis acceptance steps is also reasonable: they perform the exact accept/reject in the forward pass but omit its gradient in the backward pass (stop-gradient), citing a proposition that guarantees Newton convergence despite this discontinuity. While this is a somewhat heuristic workaround, it’s backed by theory and appears to work well in practice (the parallel MALA yields correct results). Overall, the methodology is sound and the authors seem aware of its theoretical conditions and limitations
- The authors implement their parallel MCMC algorithms in JAX on  GPU, ensuring a comparison with optimized baselines. They test the approach on multiple problems: a hierarchical Bayesian model (the 18-dimensional “eight schools” problem) using a reparameterized Gibbs sampler, a Bayesian logistic regression (BLR) with MALA and HMC on the German Credit dataset, and even a 501-dimensional item-response model (in the appendix) to demonstrate scaling to higher dimensions. For each case, the paper reports  raw speed, sample quality via Maximum Mean Discrepancy (MMD) to a ground-truth or long-run baseline. The numerical results strongly support the paper’s claims. For instance, in the BLR experiment, parallel MALA converges in only tens of Newton iterations even for very long chains. The Gibbs sampler results show a more modest but still solid gain – roughly 2× speedup over traditional sampling on the eight-schools model when chains are long. The benefit there is limited (likely by the low dimension and cheaper updates per step), but it demonstrates that even a relatively simple Gibbs chain can leverage GPU parallelism for moderate gains.

Weaknesses
- The main weaknesses of the paper lie in the limitations and scope of the proposed approach. First, while the method does achieve large speedups, it comes at the cost of substantially increased computation per iteration and memory usage. The authors explicitly note that their algorithms “trade memory and compute for time”. In scenarios where the target density or gradient is extremely expensive to evaluate, performing many parallel gradient evaluations (as this method does) might not actually be efficient – the paper acknowledges that if the model is very costly per step, those resources might be better spent on more sequential steps or independent chains.
- Secondly, the memory overhead is non-trivial. Even with the quasi-Newton diagonal approximation, parallelizing very long chains or using large batch sizes can exhaust GPU memory (the authors report out-of-memory errors for their largest tested setting, 32 parallel chains of length 64K, using the memory-hungry version). The paper’s solution – the memory-efficient Jacobian estimator – was critical to make the BLR experiments feasible, but this comes at the price of using stochastic estimates, which might in some cases slow convergence (indeed, they observed the plain quasi-Newton often took the maximum allowed iterations to converge for HMC until the block method was introduced). Thus, while the authors did mitigate this issue, scalability to very high dimensions or extremely long chains remains a concern
- One minor concern is that the analysis of convergence and conditions for quadratic speed-up are mostly cited from prior work; the paper itself does not deeply analyze when Newton’s method will converge quickly for a given MCMC (aside from empirical observation). However, given space constraints, this is understandable and the empirical results provide evidence that a few dozen Newton iterations usually suffice.
- The paper focuses on Gibbs (with continuous reparameterizable conditionals), Langevin dynamics, and HMC with fixed hyperparameters. It does not address more complex adaptive methods like NUTS or discrete-variable samplers. The authors explicitly leave extension to adaptive HMC and other MCMC variants for future work. Consequently, the immediate applicability is somewhat limited – many practical Bayesian inference tasks use NUTS or other adaptive schemes that would need non-trivial adaptation of this method. Finally, from an originality perspective, one could argue that the paper is an incremental advance in that it leverages an existing technique (parallel Newton solver from prior work) rather than inventing a wholly new sampling paradigm. The true novelty is in recognizing that this technique can be applied to MCMC and making it work for those algorithms; this is a meaningful innovation, but it’s not a completely new theory of MCMC.

---

> ### Author Rebuttal · Authors · 2025-07-30
>
> We thank the reviewer for their consideration of the paper and helpful feedback. In the following, we aim to address the reviewer’s questions about DEER, theoretical claims, and clarity of original contributions. Furthermore, we comment on the noted weaknesses, and report results applying the approach to larger-scale problems and adaptive MCMC samplers.
>
> **Elaboration on DEER algorithm (Question 1)**
>
> We agree with the reviewer that including more details about DEER to make the background self-contained will help with clarity of the paper. We have expanded the background to incorporate this feedback, notably by adding a pseudocode algorithm block that demonstrates how the DEER family of algorithms iteratively update the entire sequence of MCMC states until numerical convergence.
>
> **Theoretical Claims and Guarantees (Question 2, Weakness 3)**
>
> Our claims are typically supported by propositions from Lim et al., 2024 and Gonzalez et al., 2024. In particular, Proposition 1 from Gonzalez et al., 2024 shows that global convergence is achievable even when approximate Jacobians are substituted for the true Jacobians. Therefore, even in stochastic quasi-DEER where we have substituted in a stochastic estimate of the Jacobian, global convergence remains achievable. The sublinear scaling claims are based on the theoretical complexity of the work-efficient parallel scan ($\mathcal{O}(\log T)$ for length $T$), such that a relatively small number of parallel iterations is faster than sequential evaluation. We have clarified these statements in the paper by stating and citing the relevant propositions and assumptions made in each proposition.
>
> For claims about quadratic convergence, we rely on showing the equivalence to Newton’s method, the proof of quadratic convergence in Appendix A.3 from Lim et al., 2024, and the general theory of Newton’s method. We will add this discussion to the paper. Importantly, we note that a complete theory of the number of iterations required for DEER to converge is not available, as we do not know when the quadratic convergence criteria hold, among other factors. Deriving such a theory and connecting it to the MCMC applications shown here is an important direction of future work, but it is outside the scope of this paper. However, as the reviewer notes the empirical results provide strong evidence for convergence in tens of iterations in the considered settings.
>
> **Clarity of contributions and Novelty (Question 3, Weakness 4)**
>
> We appreciate this feedback and will aim to more clearly state the novel contributions of this paper and how it is different from previous work. We will emphasize that the bulk of work on parallelizing diffusion model sampling relies on Picard iterations, which we believe works in that setting with small step sizes but can struggle to efficiently converge in MCMC applications with larger step sizes. In fact, applying the methods proposed in this paper in diffusion model examples is a promising future direction. Next, we will elaborate on various alternative ways parallel computation has been used for MCMC, though to our knowledge a general purpose procedure for parallel evaluation of standard MCMC methods across the chain length is novel.
>
> **Scalability (Weakness 2)**
>
> The reviewer states that scalability to higher dimensions or long chains remains a concern. We agree that these are important considerations. In the following, we report an application to a higher-dimensional problem and discuss additional factors for scaling.
>
> First, we tested our approach in a sentiment classification model that predicts a binary positive or negative sentiment given a 768-dimensional LLM embedding of a text sequence. We used 1024 examples from the IMDB review dataset for this problem. We used parallel or sequential MALA to sample from the 768D posterior of this Bayesian logistic regression model. We found that parallel MALA simulated 4 chains of 4096 samples from this model about **3.66x faster** than sequential MALA (mean sequential time = 0.448s, mean parallel time = 0.122s). This importantly demonstrates feasibility of the approach in higher dimensions. We will include this result in the final version of the paper.
>
> Next, we address a few points regarding scaling. First, in practice we found that the stochastic quasi-DEER method rarely requires more iterations than the original diagonal quasi-DEER method, and the use of stochastic estimators is typically not a problem for scaling. Second, after submission we have also adopted a sliding window approach (akin to Shih et al., 2023) for scaling to longer sequences. The window can be set to the maximum sequence length supported efficiently in memory for the given batch size.
>
> **Significance of Advance and Adaptive MCMC (Weakness 4)**
>
> We focused on non-adaptive methods for this paper, as Gibbs, MALA and well-tuned HMC are useful in their own right for many inference and machine learning applications. We believe there are potential immediate applications of this work for machine learning systems that leverage Langevin dynamics, for approximate inference (i.e. via the proposed early stopping procedure), and for training algorithms that rely on MCMC sampling such as Monte Carlo EM. Furthermore, we envision that this work may open up new opportunities to apply MCMC sampling in settings where it was previously too time-consuming to run a sequential chain. Finally, the proposed stochastic quasi-DEER algorithm solves a general challenge for quasi-DEER, as computing the diagonal Jacobian can generally be time-consuming and memory intensive. The stochastic estimate can be computed in a single forward pass of the dynamics function that jointly evaluates the function at a specified input and returns the Jacobian-vector product (e.g. using `jax.jvp`). We therefore believe it will be generally useful for nonlinear sequence model training. For those reasons we believe this paper is more than an incremental advance over previous work.
>
> However, we certainly agree with the reviewer that extending the approach to adaptive MCMC samplers is an important and exciting direction of future work. We tested the feasibility of parallelizing NUTS using Blackjax’s implementation of NUTS targeting the Bayesian logistic regression posterior for the German Credit dataset. On this model, we found that our proposed methods can effectively parallelize NUTS sampling under certain conditions and the resulting sampler was faster than sequential NUTS evaluation. With a batch size of 4 chains and 1024 samples per chain, the parallel sampler was about **3.98x faster** than sequential sampling (mean sequential time = 1.139s, mean parallel time = 0.286s). These initial results, while preliminary, are a promising signal that the proposed approach can be extended to adaptive MCMC samplers such as NUTS. Furthermore, methods for dynamic HMC such as jittering the step size or number of leapfrog steps can be even more directly incorporated into our framework.

---

### Official Review · Reviewer_m4iF · 2025-07-03

**Clarity:** 2
**Significance:** 2
**Originality:** 3
**Rating:** 3
**Confidence:** 3

**Summary:**

Previous work has shown that nonlinear RNNs can be evaluated in parallel using Newton's
method (DEER) and this has been made stable and faster in subsequent work (ELK).
The current paper applies this technique to the task of drawing MCMC samples thus
enabling withing-chain parallelism.

The paper also explores the application of various standard approximations to further
bring down the computation cost to a product of the state space times the length of the
sequence (as opposed to quadratic in the size of the state space).

Further the paper demonstrates that the application of DEER (and its approximations) to
MCMC chains leads to very fast convergence. Thus we don't need a very long sequence of
states.

**Questions:**

On line 94 what does it mean to say "g <- \sigma(g) + ... "? (This looks like a recurrence?)

How big is the posterior state space in the German Credit Bayesian Logistic Regression
example?

How does this method work on problems with multimodal posteriors? Does the small number of
samples that seem to be needed for DEER also allow for efficient exploration of
multimodal posteriors.

**Ethical Concerns:**

["NO or VERY MINOR ethics concerns only"]

**Final Justification:**

The authors claim of improving over the performance of variational methods on unimodal non-gaussian posteriors is not demonstrated to a convincing degree in this paper. MCMC methods need to demonstrate excellence in multimodal posteriors which this paper doesn't attempt to demonstrate. The authors tease about some results in the rebuttal which cannot be validated. In any case, if the authors were aware of other MCMC papers with specific multimodal posteriors used for evaluations then those should have been front and center in their evaluations.

**Limitations:**

yes

**Quality:**

2

**Strengths And Weaknesses:**

## Strengths - novel application of RNN literature to MCMC

The original DEER and ELK were designred for parallelizing nonlinear RNNs. The authors
have taken that work and drawn an interesting parallel to MCMC. This appears to be
novel and allows for MCMC to leverage advances in a different area of AI.

## Strengths - fast convergence and high ESS.

The experiments do seem to suggest that the application of DEER to MCMC allows for
convergence with very few samples and these tend to have very high Effective Sample Size
(ESS). This makes the approach very promising.

## Weakness - unclear MALA algorithm

I couldn't follow the use of the stop_gradient in the MALA algorithm. I do understand
how this trick normally works, but this formulation of the MALA algorithm was not very
clear to me and Appendix A doesn't have any further explanation even though line 101
promises that appendix A would have additional details.

## Weakness - very small scale experiments

The N schools example only had an 18-dim posterior. Given the significant improvements
claimed in the paper the authors could have demonstrated on a higher dimenstional problem.

## Weakness - no demonstration of accuracy

The paper has proposed a number of approximations but none of the experiments demonstrate
that the results are actually correct.

Further none of the models have multimodal posteriors so it is unclear whether all of these
approximations will allow for efficient exploration of the posterior space.

---

> ### Author Rebuttal · Authors · 2025-07-30
>
> We thank the reviewer for their consideration of the paper and for their comments and questions. In the following, we clarify the MALA algorithm details, comment on the accuracy of the posteriors, present a new result demonstrating efficacy of the approach in a higher-dimensional problem, and address the reviewer’s question about multimodal posteriors.
>
> **MALA algorithm details**
>
> We greatly appreciate the question and feedback about our description of the MALA algorithm. There is a typo in line 94 of our writeup of the stop gradient trick that likely caused this confusion. The application of the stop gradient trick in line 94 should be written as $g \leftarrow \sigma(\tilde{g}) + \mathrm{stop gradient} (\mathbf{1}(\tilde{g}>0) - \sigma(\tilde{g}))$. Specifically, we introduce a gating variable $g$ that is binary in the forward evaluation of the function such that it implements a true, discrete accept-reject step. When computing the Jacobian, we interpolate between the current and proposed states using a sigmoid. Therefore, the ground truth sequential samples are true samples from a MALA sampler and the parallel sample sequence converges to the same sample sequence, up to numerical tolerance (in fact we tune the step size to ensure a reasonable rejection rate and observe rejected samples in practice). We have revised the manuscript with the corrected equation and improved explanation of this point.
>
> **Accuracy of algorithms**
>
> The reviewer states none of the experiments demonstrate the accuracy of the approximations. Here we hope to respectfully describe how we believe this is a misconception. Outside of the early-stopping results presented in Figure 6B, the parallel MCMC samplers converge to the ground-truth sequential samples, which are not approximate. Additionally, we put considerable effort into showing the quality of samples returned from the parallel MCMC samplers. We elaborate on both these points below.
>
> First, we note that the sequential sampler in each example is not approximate, as it generates a set of ground truth samples obtained from running a Gibbs, MALA, or HMC sampler with manually tuned hyperparameters targeting a posterior of interest (we note the Gibbs sampler actually has no hyperparameters). Each example shown targets a commonly-used posterior for evaluation of MCMC algorithms. We showed for each example how samples from the parallel version of the algorithm converge to the ground truth sequential samples up to the numerical tolerance.
>
> Second, for the MALA example we importantly used the maximum mean discrepancy (MMD) metric to evaluate the quality of the parallel and sequential samples as a function of wall-clock time. The MMD metric compared the simulated samples to a “gold-standard” set of 100K MCMC samples obtained using the No-U-Turn Sampler (NUTS). Both sequential and parallel MALA samplers achieved lower MMD values as more samples were obtained, which additionally demonstrated the accuracy of the samplers.
>
> We will amend the manuscript to make these points more clear. Furthermore, we will also illustrate how the resulting posterior mean of the MALA model provides accurate predictions for the Bayesian logistic regression model.
>
> **Dimensionality of experiments and extending to higher dimensions**
>
> We agree about the importance of testing the approach on higher-dimensional distributions, as the German Credit example has 24 features. Here, we will describe an experiment demonstrating the approach in a significantly higher-dimensional problem targeting a posterior with 768 dimensions.
>
> Specifically, we tested parallel MALA in a sentiment classification model that predicts a binary positive or negative sentiment given an LLM embedding of a text sequence. We used 1024 randomly selected examples from the IMDB Sentiment Analysis dataset for this problem. We computed a 768 dimensional embedding of each IMDB text sequence using Gemini Embedding and used a Bayesian logistic regression model to predict the sentiment from the embedding. Hence, the resulting model targets a higher-dimensional posterior with 768 dimensions.
>
> We found that parallel MALA simulated 4 chains of 4096 samples from this model about **3.66x faster** than sequential MALA (mean sequential time = 0.448s, mean parallel time = 0.122s). This importantly demonstrates feasibility of the approach in higher dimensions and we will include this result in the final version of the paper. Furthermore, while it is important to see successful scaling of the algorithm to higher dimensions, we also want to emphasize that it is not uncommon to have tens of dimensions for MCMC applications. Parallelizing MCMC for those problems is important in its own right.
>
> **Multimodal posteriors**
>
> The efficacy of this approach for multimodal posteriors is an important question, as outside of the Gibbs sampler we primarily focused on standard or hierarchical Bayesian GLMs in the submitted manuscript. First, since the parallel samplers converge to the sequential sampling states, the efficacy of exploring multimodal distributions depends on the ability of the sequential sampler to explore such distributions. However, we note that DEER is not limited to small numbers of samples. The parallelization actually can enable efficient generation of much longer MCMC sequences, which may help algorithms explore multimodal distributions.
>
> Here we demonstrate applicability of the samplers to multimodal distributions in a toy example. We tuned MALA to target a mixture of 4 Gaussians in two dimensions such that it explored all modes. Using quasi-DEER, we found parallel MALA could evaluate a 10K length sequence with 41 parallel iterations, converging to the ground truth sequential trace. We have revised the manuscript to include this example in the appendix.
>
> **Conclusion**
>
> We hope to have addressed the reviewer’s concerns regarding the MALA algorithm, accuracy of the samplers, and scalability of the algorithm to higher-dimensions, and the reviewer’s questions regarding multimodal distributions. If the reviewer has any remaining concerns or questions, we are happy to address them in the discussion phase.

---

> > ### Comment · Reviewer_m4iF · 2025-08-07
> > **Response**
> >
> > I am willing to concede a number of points to the authors, however, I cannot let go of the lack of results on multimodal posteriors. The ability of MCMC methods to explore multimodal posteriors is what separates it from more efficient methods like Variational Inference or Laplace Approximations that work very well for unimodal posteriors. Hence, a paper with MCMC in its title and which doesn't prove that it can work on multimodal posteriors doesn't seem like a worthwhile contribution.
> >
> > BTW, a mixture of Gaussians is too simplistic and also it may not even be multimodal if the peaks are close together.

---

> > > ### Author Response · Authors · 2025-08-07
> > >
> > > We thank the reviewer for carefully considering our response. We agree about the importance of demonstrating the approach on multimodal posteriors. However, it is important to note that **MCMC methods have a clear advantage over variational inference and Laplace approximation approaches for exploration of non-Gaussian posteriors, including unimodal distributions.** In fact, **our paper includes two examples** where we applied parallel MCMC methods to sample from distributions that VI and Laplace approximations would struggle to capture.
> > >
> > > The first is the Rosenbrock (banana-shaped) distribution in Figure 1. This distribution is highly-curved and neither standard variational inference nor a Laplace approximation would accurately capture this distribution. However, the parallel MCMC sampler efficiently explores the curvature of the distribution. The second is the Gibbs sampling example, where we specifically point the reviewer to the samples shown in Figure 7 of the appendix. The posterior distribution over $\tau$ is asymmetric and has a long tail. This is efficiently captured with our parallel MCMC method, whereas standard VI and Laplace approximations would not correctly capture the tails of this distribution.
> > >
> > > Next, while we agree that mixtures of Gaussians are not necessarily multimodal, **we carefully designed our mixture of Gaussians example to ensure it had multiple distinct modes.** In this case it had 4 separated modes, each of which was explored by the sampler. We recognize that a mixture of Gaussians is a synthetic example, but it is important to emphasize that they are a commonly chosen target distribution in the literature for testing MCMC sampler performance on multimodal distributions. To support this point, we refer the reviewer to two recent papers that we cite in our main text. The first is Margossian et al., 2024 [1], a journal paper on MCMC analysis, and the second is Dance et al., ICLR 2025 [2], a recent ML conference paper on vectorizing MCMC. In both papers, the primary multimodal example was a mixture of Gaussians. In general, throughout the paper we specifically chose common target distributions for testing the performance of MCMC samplers.
> > >
> > > In summary, we have demonstrated parallel MCMC simulation on multiple challenging target distributions where MCMC significantly outperforms standard variational inference and Laplace approximation methods. We would be happy to discuss this further or address any remaining concerns if the reviewer has them.
> > >
> > > [1] Margossian, Charles C., et al. "Nested ˆR: Assessing the convergence of Markov chain Monte Carlo when running many short chains." Bayesian Analysis 1.1 (2024): 1-28.
> > >
> > > [2] Dance, Hugh, et al. "Efficiently Vectorized MCMC on Modern Accelerators." Forty-second International Conference on Machine Learning (2025).

---

### Official Review · Reviewer_G15U · 2025-07-03

**Clarity:** 2
**Significance:** 4
**Originality:** 3
**Rating:** 5
**Confidence:** 3

**Summary:**

MCMC has a computational cost that is linear to the number of samples that one would like to simulate. This computational cost can be improved via parallelization. Most existing methods does so by simulating several independent Markov chains and aggregate all resulting samples. This paper takes a different approach and simulates a sequence of MCMC states (given a fixed length) by treating the sequence as the solution to a fixed-point problem, which can be solved using a parallelized form of Newton's method. The author specifies the nonlinear sequence model for a number of MCMC methods, which can be used to obtain the MCMC state sequence as the solution of corresponding fixed-point problems. It is shown that this approach can lead to improved wall-clock times compared to the traditional sequential simulation of MCMC states when the number of states generated is the same.

**Questions:**

One suggestion that I have is to expand the background section. While the author does provide all necessary citations to understand the DEER algorithm (which is a core component of the proposed methodology), the text in section 2 itself may not be detailed enough for those that have not heard of the DEER algorithm before (myself included).

More specifically, it wasn't immediately clear to me that the number (T) of MCMC states to be simulated is fixed, and that the Newton iterations are optimizing the entire sequence of T states as a whole. It may be a good idea to outline the DEER algorithm in section 2. This would help make the subsequent sections of the paper easier to read.

**Ethical Concerns:**

["NO or VERY MINOR ethics concerns only"]

**Final Justification:**

This paper looks at parallelizing MCMC from a different but interesting angle, and the authors addressed my questions during the rebuttal period. I am keeping my positive score.

**Limitations:**

Yes.

**Quality:**

3

**Strengths And Weaknesses:**

This paper is for the most part very well organized and clearly written. The problem studied here is very important as improving the computational cost of MCMC is a long-standing topic. The proposed algorithm here is easy to understand and simple to use. Compared to other methods that combine states from multiple independent MCMC instances, there is no extra care needed for aggregating samples from different chains.

---

> ### Author Rebuttal · Authors · 2025-07-30
>
> We thank the reviewer for their consideration and feedback. We discuss below how we have improved the background section to provide more details on DEER. Additionally, we report two new experiments we performed that test the approach for 1) adaptive MCMC samplers and 2) higher-dimensional problems.
>
> **Additional Background on DEER**
>
> We appreciate the suggestion to add more background on DEER to the methods, as we agree that this will improve clarity. We have revised the paper accordingly. In particular, we added a pseudocode algorithm block showing the high-level steps of this approach. This more clearly shows how the algorithm initializes a guess for the full sequence of states and iteratively updates the entire sequence using a variant of DEER, which is parallel-in-time. We have also added additional explanatory details to the text, including considerations for computing the Jacobians.
>
> **Extending approach to adaptive MCMC Samplers**
>
> An important direction of future work is extending the approach to adaptive MCMC samplers such as the No-U-Turn Sampler (NUTS). We tested the feasibility of parallelizing NUTS using Blackjax’s implementation of NUTS targeting the Bayesian logistic regression posterior for the German Credit dataset. On this model, we found that our proposed methods can effectively parallelize NUTS sampling under certain conditions and the resulting sampler was faster than sequential NUTS evaluation. With a batch size of 4 chains and 1024 samples per chain, the parallel sampler was about **3.98x faster** than sequential sampling (mean sequential time = 1.139s, mean parallel time = 0.286s). These initial results, while preliminary, are a promising signal that the proposed approach can be adapted to adaptive MCMC samplers such as NUTS.
>
> **Demonstration of approach in higher-dimensions**
>
> Finally, we also want to mention an additional experiment we performed investigating the method in a higher-dimensional problem. We tested our approach in a sentiment classification model that predicts a binary positive or negative sentiment given a 768-dimensional LLM embedding of a text sequence. We used 1024 examples from the IMDB review dataset for this problem. We used parallel or sequential MALA to sample from the 768D posterior of this Bayesian logistic regression model. We found that parallel MALA simulated 4 chains of 4096 samples from this model about **3.66x faster** than sequential MALA (mean sequential time = 0.448s, mean parallel time = 0.122s). This importantly demonstrates feasibility of the approach in higher dimensions. We will include this result in the final version of the paper.

---

> > ### Comment · Reviewer_G15U · 2025-08-09
> > **Reply to authors**
> >
> > Thank you for the reply. I would like to maintain my positive score.

---

### Official Review · Reviewer_Eern · 2025-07-07

**Clarity:** 4
**Significance:** 4
**Originality:** 4
**Rating:** 6
**Confidence:** 3

**Summary:**

Previous work on parallelising MCMC on multi-core CPUs and GPUs has focused on running multiple chains in parallel. This method takes recent work on parallel evaluation of non-linear recursions using Newton's method (DEER, quasi-DEER) and proposes a framework to run an MCMC sampler in parallel across the chain length. The authors demonstrate their approach by extending Gibbs sampling, Metropolis-adjusted Langevin, and Hamiltonian Monte Carlo. They then extend their framework to use quasi-Newton methods which reduces the wall-clock runtimes by orders of magnitude for a number of test problems.

**Questions:**

1) Do you see a clear path towards extending your framework to support adaptive MCMC samplers? If so, could you include a sentence or two about what would be required? It is not clear whether this is feasible.

**Ethical Concerns:**

["NO or VERY MINOR ethics concerns only"]

**Limitations:**

Yes

**Quality:**

4

**Strengths And Weaknesses:**

Strengths

1) Very explanation of the problem, and demonstration of their solutions effectiveness with a number of widely-used MCMC samplers.
2) Significantly reduces wall-clock time for a number of MCMC samplers by permitting the use of greater computational resources.
3) Good evaluation of the method for a number of test problems, including Bayesian Logistic Regression.

Weaknesses

1) Method cannot clearly be applied directly to adaptive MCMC samplers.
2) Speedup is less than linear relative to hardware, so 2x hardware doesn't lead to 2x performance improvement. Although this is mitigated through the availability of GPUs.

---

> ### Author Rebuttal · Authors · 2025-07-30
>
> We thank the reviewer for their consideration and comments. In the following, we discuss the feasibility of extending the framework to adaptive MCMC samplers such as the No-U-Turn Sampler (NUTS). We also comment on the relationship between the wall-clock speedup and hardware and report results on applying the framework to a higher-dimensional problem.
>
> **Adaptive MCMC Samplers (Question 1, Weakness 1)**
>
> Extending the proposed framework to adaptive MCMC samplers is an important and exciting direction of future work. Here, we discuss considerations for variations of adaptive MCMC and also report the results of an initial test of applying the approach to parallelizing NUTS targeting the Bayesian logistic regression posterior on the German credit dataset. We will include the following discussion of adaptive MCMC methods in the final version of the paper.
>
> NUTS introduces additional control flow for building the set of candidate points. As the proposed algorithms in this paper leverage the Jacobian of the update step to propagate information, this potentially complicates extending the approach to NUTS. Nonetheless, we aimed to quickly test the feasibility of this approach. We used the Blackjax package’s NUTS implementation [Cabezas et al., 2024], which is compatible with a forward-mode Jacobian-vector product out of the box because it uses `jax.lax.cond`. This propagates gradient information along the evaluated branch of the algorithm but notably does not differentiate through the predicate that determines which branch to evaluate. We did not modify the code to attempt to improve the Jacobian signal (for example, by substituting in differential relaxations for Jacobian computation).
>
> On the Bayesian logistic regression model for the German credit dataset, we found that the approach can effectively parallelize NUTS sampling under certain conditions and the resulting sampler was faster than sequential NUTS evaluation. On a GPU with a batch size of 4 chains and 1024 samples per chain, the parallel sampler was approximately **3.98x faster** than sequential sampling (mean sequential time = 1.139s, mean parallel time = 0.286s). These initial results, while preliminary, are a promising signal that the proposed approach can be adapted to NUTS.
>
> Aside from NUTS, a simpler approach to avoid potential HMC pathologies with a fixed step size and trajectory length is to randomly jitter either the step sizes or number of leapfrog steps across sample iterations (sometimes called Dynamic HMC). While we have not tested this, these methods are straightforward to incorporate into our approach. In particular, the jittered step sizes or number of leapfrog steps can be presampled and wrapped up into the time-varying function $f_t$. Notably, ChEES-HMC [Hoffman et al., 2021] is an adaptive HMC algorithm that uses dynamic trajectory lengths after tuning parameters during a warm-up phase, and our approach could be applied using the tuned parameters after an initial warm-up.
>
> **Relationship between wall-clock speedup and hardware (Weakness 2)**
>
> Here we discuss the theoretical efficiency of the algorithm and how it scales with additional hardware resources, considering the reviewer’s comment on wall-clock speedup scaling with hardware. We focus on the work-efficient parallel scan used to evaluate the linear recursion. If $P$ is the number of available parallel processors, then the time complexity for evaluating a length $P$ sequence is $\mathcal{O}(\log P)$. If the number of parallel processors is doubled to $2P$, then we can then evaluate a length $2P$ sequence in $\mathcal{O}(\log 2P)$ time. This in fact implies a beneficial theoretical scaling of the algorithm with additional hardware for sequence length, where for $P>2$ doubling the processors enables evaluating twice the length of a sequence in less than twice the amount of time.
>
> **Demonstration of approach in higher-dimensions**
>
> Finally, we report an additional experiment we performed investigating the method in a higher-dimensional problem. We tested our approach in a sentiment classification model that predicts a binary positive or negative sentiment given a 768-dimensional LLM embedding of a text sequence. We used 1024 examples from the IMDB review dataset for this problem. We used parallel or sequential MALA to sample from the 768D posterior of this Bayesian logistic regression model. We found that parallel MALA simulated 4 chains of 4096 samples from this model about **3.66x faster** than sequential MALA (mean sequential time = 0.448s, mean parallel time = 0.122s). This importantly demonstrates feasibility of the approach in higher dimensions. We will include this result in the final version of the paper.

---

> > ### Comment · Reviewer_Eern · 2025-08-06
> >
> > Thank you for the detailed rebuttal. I do not change my rating.

---

### Note · Authors · 2025-08-13

Thank you to the reviewers for their helpful comments and discussion of the paper. Our revised paper has been strengthened through this feedback.

Overall, our paper presents methods to parallelize MCMC sampling across the sequence length. We applied this to parallelize Gibbs, MALA, and HMC sampling in multiple examples including a Bayesian regression model, a hierarchical linear model, and a distribution with high-curvature. The reviewers generally found the work to be original and significant. Reviewer hixK even stated that “this breakthrough transforms MCMC” from a sequential process to a parallelizable problem, “opening new possibilities for scaling Bayesian inference.”

The reviewers also noted specific comments and questions, which we addressed in our individual responses. Some of the primary points included:
1. Reviewers suggested elaborating on the DEER method in our background. We have revised the paper with a detailed description of DEER and an algorithm block that highlights the key steps of the DEER method.
2. Reviewers asked about how the algorithm scales to higher-dimensional or larger problems. We have added a 768-dimensional sentiment classification example in our revision, demonstrating the effectiveness of our method on larger MCMC problems.
3. Reviewer m4iF highlighted the importance of adding a multimodal distribution example. We have revised the paper with a highly multimodal mixture of Gaussians experiment, a common test of MCMC algorithms’ ability to explore multimodal distributions. We show our approach can effectively parallelize a MALA sampler that traverses well-separated modes of the distribution.

Overall, these recommendations and additions have significantly improved the paper, and we believe our work will be of great interest to the NeurIPS community. Again, we thank the reviewers for their engagement and constructive feedback and the AC for facilitating the review process.

---

### Decision · Program_Chairs · 2025-09-17

**Decision:**

Accept (poster)

**Comment:**

Standard MCMC is inherently sequential, with cost scaling linearly in the number of samples, which makes it poorly suited to parallel hardware. Prior work mostly runs many chains in parallel (ie. embarrassingly parallel setting), leaving the sequential bottleneck untouched. This paper asks whether a single chain can be parallelized across its length, turning an O(T) process into something closer to O(log T). This direction could make MCMC practical at scales where sequential simulation is too slow.

The main contribution is to show that Gibbs, MALA, and HMC can be written as fixed-point recurrences and evaluated in parallel using Newton-type methods, building on the DEER (non-linear Differential Equation as fixed point itERation) framework. The authors propose practical variants such as diagonal and stochastic Jacobian approximations and block updates for HMC, which reduce memory and computation and make the method usable on GPUs. They also show how the Metropolis step can be handled by exact evaluation in the forward pass with gradients omitted in the backward pass. Experiments demonstrate significant speedups, including on higher-dimensional problems. The novelty is in adapting DEER-style methods to MCMC and making them practical for standard samplers.

During rebuttal, reviewers asked for more details, scaling to higher dimensions, treatment of adaptive samplers, the accept–reject step, and performance on multimodal targets. The authors added a clearer description of the method, fixed a presentation error in MALA, and reported new experiment. These additions convinced two reviewers and softened concerns of a third, though one remained unconvinced about multimodal performance and theoretical depth.

Overall, there is agreement that the idea is novel and tackles a long-standing limitation of MCMC. While theory remain incomplete, the authors strengthened the paper and showed feasibility on challenging problems. The work opens a promising research direction.